# RE-Searcher: Robust Agentic Search with Goal-oriented Planning and Self-reflection

## Abstract

Large language models (LLMs) excel at knowledge-intensive question answering and reasoning, yet their real-world deployment remains constrained by knowledge cutoff, hallucination, and limited interaction modalities. Augmenting LLMs with external search tools helps alleviate these issues, but it also exposes agents to a complex search environment in which small, plausible variations in query formulation can steer reasoning into unproductive trajectories and amplify errors. We present a systematic analysis that quantifies how environmental complexity induces fragile search behaviors and, in turn, degrades overall performance. To address this challenge, we propose a simple yet effective approach to instantiate a search agent, RE-Searcher. During search, RE-Searcher explicitly articulates a concrete search goal and subsequently reflects on whether the retrieved evidence satisfies that goal. This combination of goal-oriented planning and self-reflection enables RE-Searcher to resist spurious cues in complex search environments and perform robust search. Extensive experiments show that our method improves search accuracy and achieves state-of-the-art results. Perturbation studies further demonstrate substantial resilience to noisy or misleading external signals, mitigating the fragility of the search process. We believe these findings offer practical guidance for integrating LLM-powered agents into more complex interactive environments and enabling more autonomous decision-making.

## 1 Introduction

Large language models (LLMs) have demonstrated remarkable performance in knowledge-intensive question answering and logical reasoning tasks (Shao et al., 2024; Li et al., 2025a; Minaee et al., 2024), and have gradually been deployed in real-world applications. Nevertheless, their further development remains constrained by several limitations: (1) **Knowledge cutoff**: model knowledge is restricted to the static pre-training corpus and cannot be updated in real time (Shah et al., 2025; Cheng et al., 2024); (2) **Hallucination**: as probabilistic generators, LLMs inevitably produce content that is inconsistent with factual knowledge or user intent (Ji et al., 2023; Huang et al., 2025; Tonmoy et al., 2024); (3) **Interaction constraint**: models typically interact in a conversational form, restricting their capacity to perform more complex tasks (Schick et al., 2023; Yao et al., 2023). These challenges substantially limit the applicability of LLMs in open and dynamic real-world scenarios.

Recent research has sought to overcome these limitations by augmenting LLMs with external search tools, thereby constructing *search agents* (Jin et al., 2025; Zheng et al., 2025; Wang et al., 2025b; Hao et al., 2025). By leveraging retrieval during response generation, such agents can extend the knowledge boundary of LLMs, alleviate hallucination, and enable more diverse downstream applications. However, while the search environment can enrich the information accessible to models, they can also introduce misleading evidence, resulting in degraded or erroneous response. In fact, as shown in Section 2, our preliminary analysis shows that the complexity of the search environment can lead to fragile interactions, which in turn amplify model errors and ultimately diminish task performance. A simple illustrative case is presented in Fig. 1. When presented with the same query, the search agent issued two different sets of search keywords across two independent trials. Although both keyword choices were semantically reasonable, the retrieved results diverged dramatically. The erroneous trajectory (left) failed to yield useful information, and subsequent refinements along this trajectory could not recover the correct answer. By contrast, the correct trajectory (right) quickly

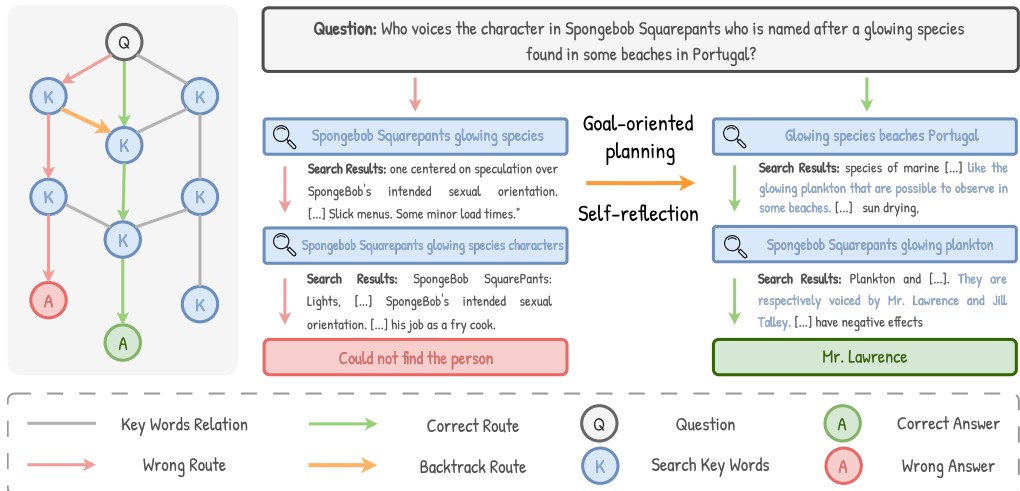

Figure 1: A search path can be viewed as a sample from the keyword graph. When receiving the same query, the search agent generates two distinct sets of keywords during two independent experiments. Although both sets of keywords are semantically sound, the retrieved results differed dramatically. Our RE-Searcher, a search agent endowed with **goal-oriented planning** and **self-reflection** (orange arrow), can recover from such missteps and return to the correct trajectory, thereby enabling robust search behavior.

retrieved the keyword "plankton" enabling the agent to find the correct answer in the second search step.

Such variability and fragility of the search process pose considerable challenges for deploying LLMs in realistic settings. In contrast, humans are remarkably robust when operating under uncertain and dynamic conditions. Prior to executing a task, humans typically form explicit expectations of the desired outcome; after completion, they engage in reflection, evaluating whether the result meets expectations before deciding on subsequent actions. This process of **goal-oriented planning** and **self-reflection** enables humans to adapt flexibly to environmental complexity.

Inspired by this cognitive paradigm, we build a search agent, **RE-Searcher**, that integrates goal-oriented planning with self-reflection. Specifically, in the search process, the agent is required to explicitly articulate its search goal and subsequently reflect on the quality of retrieved results. Our experiments demonstrate that this approach not only achieves state-of-the-art (SOTA) performance in search tasks but also substantially improves robustness. Further perturbation experiments reveal that our method enhances resilience to noisy or misleading external signals, thereby offering stronger adaptability to real-world, dynamic environments. Our contributions are listed below:

- We present a systematic analysis and quantification of how environmental complexity affects agent performance, underscoring the necessity of robustness for reliable deployment.
- We introduce a novel search agent, **RE-Searcher**, that combines goal-oriented planning with self-reflection to mitigate the impact of noisy search results and correct potentially biased trajectories, showcasing a simple yet effective approach to achieving robust search performance.
- Extensive experiments demonstrate that RE-Searcher improves search accuracy and robustness; perturbation analyses further validate the significant gains in resilience against external noise.

## 2 PRELIMINARY ANALYSIS

The practical application of search agents is severely hampered by a significant instability in their outputs for search and question-answering. In this section, we begin by quantifying this stochasticity, and then leverage our findings to propose a simple but effective methodology aimed at enhancing the agents' overall performance and robustness.[1]

---

[1]We present the main results and our analysis here. Full experimental details are available in Section A.2.

## 2.1 STOCHASTICITY OF SEARCH AGENT'S OUTPUTS

To quantify the output instability, we evaluated search agents built upon various models. Each agent performed inference twice on an identical QA dataset. We classify questions as *always right* if correctly answered in both runs, and as *random right* if correct in only one. As illustrated in Fig. 2, GPT-4o (Hurst et al., 2024), with its pre-trained tool-use capabilities (OpenAI, 2025), maintains a low, acceptable proportion of *random right* outcomes. Conversely, Qwen2.5 (Qwen et al., 2025), which lacks this prior training, exhibits a *random right* ratio that rivals or even surpasses its *always right* ratio. This highlights a critical model instability that fundamentally limits the model's achievable performance.

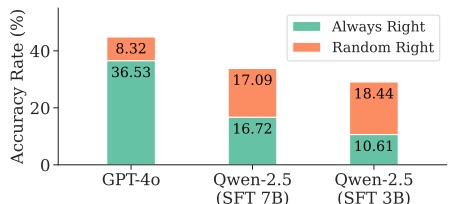

Figure 2: Accuracy rate of search agents based on different models. *always right* is the fraction of instances where all attempts are correct; *random right* is the fraction where at least one attempt is correct

## 2.2 FRAGILITY OF THE SEARCH PROCESS

Analyzing the search trajectories reveals a critical vulnerability: minuscule differences in search queries often lead to correct trajectories and incorrect ones. A single-word change in a query—such as a **synonym substitution**, **keyword addition**, or **keyword deletion**—can trigger drastically different results from the search engine. To demonstrate this, we applied these three types of micro-perturbations to search queries and measured the cosine similarity of the search results before and after. As shown in Fig. 3, even these subtle changes frequently cause a sharp decline in semantic similarity, with many results dropping below a 0.6 threshold.

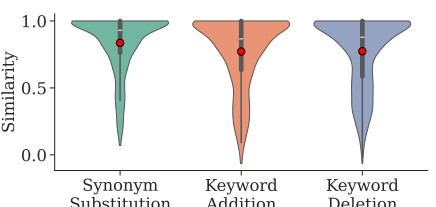

Figure 3: Cosine similarity of the search results obtained from queries before and after perturbation; the red dot indicates the mean similarity.

The complexity of search environment, therefore, acts as an amplifier for the agent's inherent stochasticity, often derailing its reasoning process towards erroneous conclusions. While a powerful model like GPT-4o can recover from such misleading signals, this underscores a general principle: an agent's ability to maintain a high-level goal and continuously self-reflect is paramount for robust performance. Motivated by this insight, our work focuses on explicitly training agents for **goal-oriented planning** and **self-reflection**. This equips them with the resilience needed to counteract the error amplification from the complex search environment.

## 3 METHODOLOGY

Table 1: Chat Template for RE-Searcher, when the model answers questions, it needs to think, plan, search, and reflect to ensure the robustness of the search path.

---

As an expert researcher, provide precise answers to the given question. When new information arrives, first reason within `<think>` and `</think>` tags to analyze the question and determine search keywords. Each search must include a clear `<goal>` specifying the information you aim to find, along with `<query>` items combining initial questions with collected information (e.g., `<search> <query>` QUERY `</query> <goal>` GOAL `</goal> </search>`). After receiving search results in `<learnings></learnings>` tags, reflect on whether they meet your goal using `<think>` for analysis, then explicitly state the outcome in `<reflect>` True/False `</reflect>` (True = goal met, False = needs refinement). If knowledge gaps exist, perform up to five iterative searches with refined goals/queries. When sufficient information is obtained, present the final answer within `<answer> </answer>` tags.

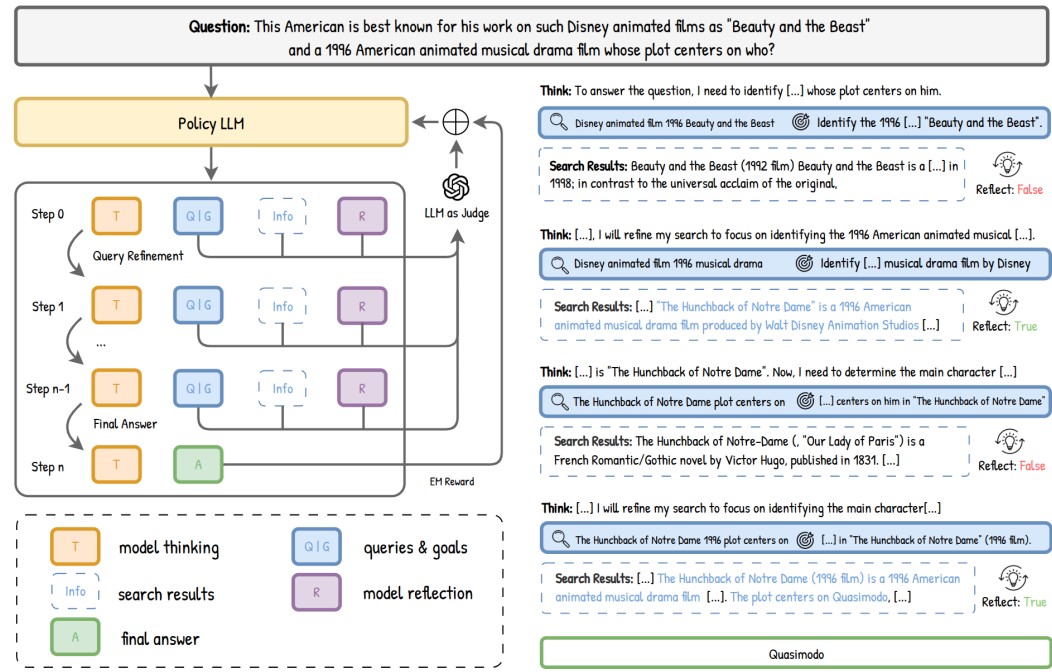

Figure 4: Illustration of the proposed training methods. Left: The model is required to explicitly plan its search goals during the search process and reflect on the results after obtaining them. An external LLM monitors the training model's reflection results to ensure that its judgments are correct. Right: The search trajectory made by the trained agentic model shows the correct reflection and goal planning.

To enhance model robustness in complex search environments that often lead to fragile interactions, we aim to equip the agent with *goal-oriented planning* and *reflection* capabilities. As illustrated in Fig. 4, during the training phase, the model is explicitly prompted to perform goal-oriented planning and reflection. Furthermore, an advanced LLM is employed to guide the model's reflective outputs. The resulting supervisory signal is then fed back to the primary model to refine its reflection accuracy.

### 3.1 EXPLICIT SEARCHING WITH GOAL-REFLECTION BEHAVIOR

To enable the model to perform explicit goal-oriented planning and self-reflection (hereafter referred to as the goal-reflection mechanism), we employ a structured generation template, as depicted in Table 1, to constrain the model's output to one of three discrete actions at each turn: **Search**, **Reflect**, or **Answer**. Each action is preceded by a "thought" process, where the model generates its rationale to ensure the subsequent output is coherent and well-founded. The **Search** action is executed as follows: the model first analyzes the initial question and the information gathered thus far to formulate a specific search *goal* and a corresponding *query*. A search engine then executes this

---

**Algorithm 1** Iterative Search and Reflection

**Require:** User question $Q$
1: **Initialize:** Context $\mathcal{C} \leftarrow \{Q\}$, $\mathcal{G}_{\text{pending}} \leftarrow \emptyset$, $\mathcal{G}_{\text{completed}} \leftarrow \emptyset$
2: Generate an initial search goal based on the input question $Q$ and add it to $\mathcal{G}_{\text{pending}}$.
3: **while** $\mathcal{G}_{\text{pending}} \neq \emptyset$ **do**
4:     Get current goal $g_{\text{current}}$ from $\mathcal{G}_{\text{pending}}$
5:     is_goal_met $\leftarrow$ FALSE
6:     **while** NOT is_goal_met **do**
7:         Generate query $q$ based on $g_{\text{current}}$ and context $\mathcal{C}$.
8:         Retrieve results $R \leftarrow \text{SearchEngine}(q)$.
9:         Update context: $\mathcal{C} \leftarrow \mathcal{C} \cup \{R\}$.
10:        Generate judgment $J \leftarrow \text{Reflect}(R, g_{\text{current}})$.
11:        **if** $J = \text{TRUE}$ **then**
12:           is_goal_met $\leftarrow$ TRUE
13:           Move $g_{\text{current}}$ from $\mathcal{G}_{\text{pending}}$ to $\mathcal{G}_{\text{completed}}$.
14:           Identify a new search goal $g_{\text{new}}$ based on $\mathcal{C}$.
15:           $\mathcal{G}_{\text{pending}} \leftarrow \mathcal{G}_{\text{pending}} \cup \{g_{\text{new}}\}$.
16:        **end if**
17:     **end while**
18: **end while**
19: Generate final answer $A$ based on the complete context $\mathcal{C}$.
20: **return** $A$

---

*'query'* and returns the results. During the **Reflect** action, the model evaluates whether the retrieved search results align with the stated *goal*. If the goal is met, the model confirms this with a *TRUE* judgment and proceeds to formulate a new search *goal* and *query*. Conversely, if the results are unsatisfactory, the model refines the *query* and re-initiates the search process to fulfill the original goal. Finally, once all necessary information has been gathered and all sub-goals are satisfied, the model transitions to the **Answer** action, synthesizing the collected evidence to produce the final response to the user's question. The full search process is shown in Algorithm 1.

To ensure the model adheres to the required output format during training, we construct a small set of chain-of-thought (CoT) interaction trajectories (approximately 1K) as a warm-up. We build an LLM agent based on a strong instruction-following model (GPT-4o) to generate interactions that conform to the above protocol, including the thought process, search steps, reflection, and final answer. These data are then used to fine-tune the base model, enabling it to produce outputs in the desired format.

## 3.2 GRPO WITH SEARCH ENGINE

The use of reinforcement learning algorithms to improve the search capabilities of models has been widely validated (Li et al., 2025b; Wang et al., 2025b; Hao et al., 2025). In this work, to mitigate the demand for computational resources, we employ Group Relative Policy Optimization (GRPO) (Shao et al., 2024) to train the model's search and reflection abilities. For each input question $x$ in GRPO, a group of $G$ rollout trajectories, denoted as $\tau = \{y_i\}_{i=1}^{G}$, is generated using the preceding policy $\pi_{old}$, the current policy model $\pi_\theta$ is subsequently optimized by maximizing the objective function:

$$\mathcal{L}(\theta) = \mathbb{E}_{\substack{x \sim \mathcal{D}, \{y_i\}_{i=1}^{G} \\ y_i \sim \pi_{old}(\cdot|x)}} \left[ \frac{1}{G} \sum_{i=1}^{G} \min\left(r_i(\theta)A_i, \text{clip}(r_i(\theta), 1-\epsilon, 1+\epsilon)A_i\right) - \beta \mathbb{D}_{KL}[\pi_\theta || \pi_{ref}] \right] \quad (1)$$

where $\pi_{ref}$ denotes reference model, $r_i(\theta) = \frac{\pi_\theta(y_i|x)}{\pi_{old}(y_i|x)}$. $\epsilon$ and $\beta$ are hyperparameter. $A_i$ represents the advantage, computed based on the relative rewards (which will be mentioned in Section 3.3) of outputs within each group. As mentioned in Section 3.1, in each rollout, the model will take search actions using `<search></search>` tags, and the retrieved tokens that are tagged by `<learnings></learnings>` will be masked when calculating the loss.

## 3.3 REFLECTION SUPERVISION THROUGH LLM AS JUDGE

After the warm-up phase, the model has learned to output in the desired format to some extent. To further enforce the correct format during the reinforcement learning stage, we integrate format constraints with the factual reward. Specifically, the output trajectory is encouraged to continuously include the actions of search and reflection, with the final action being the answer. Following the method in Jin et al. (2025), we combine the format reward with the factual reward as follows:

$$r_{em\_format} = \begin{cases} 0.8 + 0.2 \cdot \text{FM}(\tau_{pred}), & \text{if EM}(a_{\text{pred}}, a_{\text{gt}}) = 1, \\ 0.2 \cdot \text{FM}(\tau_{pred}), & \text{if EM}(a_{\text{pred}}, a_{\text{gt}}) = 0. \end{cases} \quad (2)$$

where EM is the exact match function and FM evaluates whether the predicted trajectory $\tau_{pred}$ follows the required output format. $a_{pred}$ and $a_{gt}$ denote the predicted and ground-truth answers, respectively.

We further employ model-based evaluation, i.e., an LLM as a judge to guide the model's reflection process. Specifically, we prompt GPT-4o-mini with a triple input, comprising the search goal, the search result, and the judgment, to evaluate whether the model's reflection judgment is correct. The reflection reward is weighted and added to the factual reward with format constraints for the final reward:

$$r = r_{em\_format} + \sum_i 0.1 * \text{MBE}(g_i, s_i, v_i) \quad (3)$$

where MBE denotes the model-based evaluation. $(g_i, s_i, v_i)$ is the search goal, the search result, and the judgment for the $i$-th search action.

Table 2: Exact Match (EM) metrics on question-answering tasks. The best performance is set in **bold**. Our RE-Searcher outperforms all baselines across most in/out-of-domain datasets using both Qwen2.5-3B-Instruct and Qwen2.5-7B-Instruct as base model.

| Methods | In domain | | Out of domain | | | | | Avg. |
|---|---|---|---|---|---|---|---|---|
| | NQ | HotpotQA | TriviaQA | PopQA | 2wiki | Musique | Bamboogle | |
| **Qwen2.5-3B** | | | | | | | | |
| Direct Inference | 0.106 | 0.149 | 0.288 | 0.108 | 0.244 | 0.020 | 0.024 | 0.134 |
| CoT | 0.023 | 0.021 | 0.032 | 0.005 | 0.021 | 0.002 | 0.000 | 0.015 |
| IRCoT | 0.111 | 0.164 | 0.312 | 0.200 | 0.171 | 0.067 | 0.240 | 0.181 |
| Search-o1 | 0.238 | 0.221 | 0.472 | 0.262 | 0.218 | 0.054 | 0.320 | 0.255 |
| RAG | 0.348 | 0.255 | 0.544 | 0.387 | 0.226 | 0.047 | 0.080 | 0.270 |
| SFT | 0.249 | 0.186 | 0.292 | 0.104 | 0.248 | 0.044 | 0.112 | 0.176 |
| R1-base | 0.226 | 0.201 | 0.455 | 0.173 | 0.268 | 0.055 | 0.224 | 0.229 |
| R1-instruct | 0.210 | 0.208 | 0.449 | 0.171 | 0.275 | 0.060 | 0.192 | 0.224 |
| Search-R1-base | 0.406 | 0.284 | 0.587 | 0.435 | 0.273 | 0.049 | 0.088 | 0.303 |
| Search-R1-instruct | 0.341 | 0.324 | 0.545 | 0.378 | 0.319 | 0.103 | 0.264 | 0.325 |
| $O^2$-Searcher | **0.444** | 0.388 | 0.597 | 0.429 | 0.374 | 0.160 | 0.344 | 0.391 |
| ZeroSearch-base | 0.430 | 0.338 | **0.616** | 0.414 | 0.346 | 0.130 | 0.139 | 0.345 |
| ZeroSearch-instruct | 0.414 | 0.274 | 0.574 | **0.448** | 0.300 | 0.098 | 0.111 | 0.317 |
| OTC | **0.444** | 0.365 | 0.608 | 0.441 | 0.341 | 0.124 | 0.266 | 0.370 |
| RE-Searcher (ours) | 0.419 | **0.404** | 0.600 | 0.416 | **0.420** | **0.166** | **0.408** | **0.405** |
| **Qwen2.5-7B** | | | | | | | | |
| Direct Inference | 0.134 | 0.183 | 0.408 | 0.140 | 0.250 | 0.031 | 0.120 | 0.181 |
| CoT | 0.048 | 0.092 | 0.185 | 0.054 | 0.111 | 0.022 | 0.232 | 0.106 |
| IRCoT | 0.224 | 0.133 | 0.478 | 0.301 | 0.149 | 0.072 | 0.224 | 0.226 |
| Search-o1 | 0.151 | 0.187 | 0.443 | 0.131 | 0.176 | 0.058 | 0.296 | 0.206 |
| RAG | 0.349 | 0.299 | 0.585 | 0.392 | 0.235 | 0.058 | 0.208 | 0.304 |
| SFT | 0.318 | 0.217 | 0.354 | 0.121 | 0.259 | 0.066 | 0.112 | 0.207 |
| R1-base | 0.297 | 0.242 | 0.539 | 0.202 | 0.273 | 0.083 | 0.296 | 0.276 |
| R1-instruct | 0.270 | 0.237 | 0.537 | 0.199 | 0.292 | 0.072 | 0.293 | 0.271 |
| Search-R1-base | 0.480 | 0.433 | 0.638 | 0.457 | 0.382 | **0.196** | 0.432 | 0.431 |
| Search-R1-instruct | 0.393 | 0.370 | 0.610 | 0.397 | 0.414 | 0.146 | 0.368 | 0.385 |
| ZeroSearch-base | 0.424 | 0.320 | **0.664** | **0.604** | 0.340 | 0.180 | 0.333 | 0.409 |
| ZeroSearch-instruct | 0.436 | 0.346 | 0.652 | 0.488 | 0.352 | 0.184 | 0.278 | 0.391 |
| OTC | 0.444 | 0.366 | 0.597 | 0.431 | 0.311 | 0.130 | 0.250 | 0.361 |
| RE-Searcher (ours) | **0.453** | **0.437** | 0.638 | 0.454 | **0.473** | 0.194 | **0.496** | **0.449** |

## 4 EXPERIMENTS

In this section, we design and conduct a series of experiments to answer the following key research questions (RQs):

- **RQ1**: Does the goal-reflection mechanism improve problem-solving capabilities in search tasks? (Section 4.2)
- **RQ2**: To what extent does the goal-reflection mechanism mitigate the negative impacts of search fragility? (Section 4.3)
- **RQ3**: How much does the proposed framework enhance the model's robustness against external disturbances? (Section 4.4)
- **RQ4**: Does the goal-reflection mechanism lead to unnecessary over-reasoning, or does it perform adaptive search based on the difficulty of the problem? (Section 4.5)

### 4.1 IMPLEMENTATION DETAILS

**Setup**. We adopt Qwen2.5-3B-Instruct and Qwen2.5-7B-Instruct (Yang et al., 2024) as the backbone models of our proposed RE-Searcher. For the cold start stage, we utilize the Adam optimizer with an initial learning rate of $1 \times 10^{-5}$ and a warm-up ratio of 0.1. This stage is conducted on 8 A100 GPUs for 2 epochs. During the RL training stage, we employ the Verl framework [2]. We optimize the policy model using the GRPO algorithm. At each training step on 8 A100 GPUs, we sample a batch of 64 prompts, generating 8 rollout trajectories for each. The model is updated with the Adam optimizer at a learning rate of $1 \times 10^{-6}$. For GRPO, we set the KL divergence regularization coefficient $\beta$ to 0.001 and the clip ratio $\epsilon$ to 0.2. The maximum sequence length is configured to be $10k$ tokens, while retrieved content is restricted to $2k$ tokens, and the maximum number of action

---

[2]https://github.com/volcengine/verl

Table 3: Ablation on reflection reward on multi-hop datasets. The validation samples are selected with the protocol of Zheng et al. (2025).

| Variants | HotpotQA | | 2wiki | | Musique | | Bamboogle | |
|---|---|---|---|---|---|---|---|---|
| | EM | F1 | EM | F1 | EM | F1 | EM | F1 |
| w/o reflection reward | 0.420 | 0.545 | 0.414 | 0.487 | 0.183 | 0.270 | 0.411 | 0.533 |
| w/ reflection reward | 0.431 (**+0.011**) | 0.544 (-0.001) | 0.476 (**+0.062**) | 0.549 (**+0.062**) | 0.197 (**+0.014**) | 0.290 (**+0.020**) | 0.480 (**+0.069**) | 0.578 (**+0.045**) |

Table 4: Ablation on reward components on in-domain and out-of-domain datasets. The validation samples are selected with the protocol of Zheng et al. (2025).

| Variants | In domain | Out of domain | AVG. |
|---|---|---|---|
| baseline | 0.403 | 0.395 | 0.397 |
| w/o format reward | 0.397 (**-0.006**) | 0.388 (**-0.007**) | 0.390 (**-0.007**) |
| w/o reflection reward | 0.396 (**-0.007**) | 0.387 (**-0.008**) | 0.389 (**-0.008**) |

steps is 11. To accelerate LLM rollouts, we leverage vLLM [3] with a tensor parallel size of 1 and a GPU memory utilization ratio of 0.85. For rollout sampling, we use a temperature of 1.0 and a top-$p$ value of 1.0.

**Datasets.** We assess our proposed RE-Searcher on both in-domain and out-of-domain datasets. The models are trained on in-domain datasets, including NQ (Kwiatkowski et al., 2019) and Hot-potQA (Yang et al., 2018), while the out-of-domain datasets encompass TriviaQA (Joshi et al., 2017), PopQA (Mallen et al., 2022), 2WikiMultiHopQA (Ho et al., 2020), Musique (Trivedi et al., 2022b), and Bamboogle (Press et al., 2022). In total, these validation tests involve 51,953 questions with corresponding ground-truth answers.

**Baselines.** We follow the setting of Search-R1 (Jin et al., 2025) and compare our RE-Searcher against two categories of methods: (1) CoT-based approaches, including CoT (Wei et al., 2022), RAG (Lewis et al., 2020), IRCoT (Trivedi et al., 2022a), and Search-o1 (Li et al., 2025b). These methods leverage Chain-of-Thought reasoning either for direct inference or in combination with Retrieval-Augmented Generation (RAG). (2) Train-based methods, such as Supervised Fine-Tuning (SFT) (Chung et al., 2024), DeepSeek-R1 (Guo et al., 2025), Search-R1 (Jin et al., 2025), Ze-roSearch (Sun et al., 2025), $O^2$-Searcher (Mei et al., 2025), and OTC (Wang et al., 2025a). SFT and DeepSeek-R1 perform reasoning and answer steps without a search engine, whereas other methods incorporate a local search engine.

**Metrics.** The Exact Match (EM) and F1 metrics are applied, following Yu et al. (2024); Jin et al. (2025).

## 4.2 EFFECTIVENESS OF THE GOAL-REFLECTION MECHANISM

### 4.2.1 IMPROVEMENT OF SEARCHING ABILITY

We conducted a comprehensive evaluation of RE-Searcher on both in-domain and out-of-domain tasks, with detailed results presented in Table 2. The findings clearly indicate that our method establishes a new state-of-the-art, outperforming all baseline methods across both the 7B and 3B model scales. Using the Qwen2.5-7B-instruct model as the backbone, RE-Searcher achieves the highest average EM score of 0.449, surpassing all other approaches. Notably, it secures top performance on both in-domain datasets, NQ and HotpotQA, demonstrating its proficiency on familiar tasks. Furthermore, it shows exceptional generalization to out-of-domain datasets, achieving the best scores on 2WikiMultiHopQA and Bamboogle. Compared to recent RL-based baselines, such as Search-R1 and ZeroSearch, our method provides a significant improvement in average performance, underscoring the effectiveness of our approach. Notably, the performance gains of RE-Searcher on single-hop datasets (NQ, TriviaQA, and PopQA) are less pronounced than on multi-hop datasets and are even lower than those of ZeroSearch. This is mainly because, in single-hop settings, the necessary information is often largely contained in the question itself, so complex reasoning is rarely required,

---
[3]https://github.com/vllm-project/vllm

leaving limited room for RE-Searcher to distinguish itself. By contrast, ZeroSearch uses an LLM pre-trained on search corpora as its retrieval engine, providing stronger semantic matching and thus better performance on single-hop benchmarks.

To validate the scalability and efficiency of our method, we also evaluated it on the smaller Qwen2.5-3B-instruct model. The results reinforce our claims, as RE-Searcher again achieves the highest average EM score of 0.405, outperforming competitive methods like $O^2$-Searcher and OTC. This consistent superiority highlights the scalability and robust effectiveness of our approach. Fig. 4 shows a search trajectory of RE-Searcher. The model plans the search goal for each search and reflects on whether the retrieved content meets the requirements. During the third search, the search engine incorrectly returned information about a novel with the same title. Through reflection, the model simply modified a single keyword and obtained the correct result.

### 4.2.2 ANALYSIS ON REFLECTION REWARD

We analyze the impact of the reflection reward on training dynamics. As illustrated in Fig. 5, the model trained without this reward exhibits a reflection score that hovers around 0.5. This indicates a near-random judgment on the consistency between the retrieved information and the search goal, underscoring the importance of the explicit guidance provided by the LLM-as-judge. In contrast, with the reflection reward, the score stabilizes at a higher value, demonstrating that the model learns a consistent and effective reflection policy. These training dynamics are corroborated by quantitative results on the validation set. As shown in Table 3 and

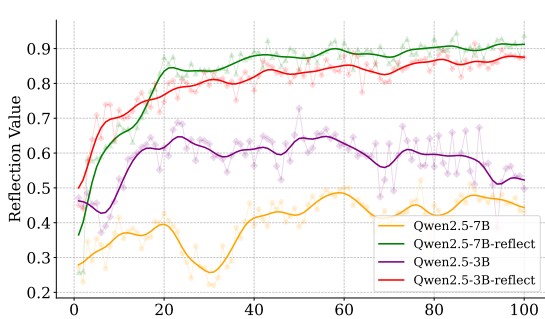

Figure 5: The training dynamics of the reflection value of different models.

Table 4, removing the reflection reward of RE-Searcher (7B) and (3B), respectively, leads to a consistent performance drop across both in-domain and out-of-domain datasets, as well as all evaluated multi-hop datasets. Conversely, its inclusion yields significant improvements, particularly on the more challenging 2wiki (+0.062 in both EM and F1) and Bamboogle (+0.069 in EM and +0.045 in F1) datasets. While the gains on Musique are more modest, they remain consistently positive across both metrics.

To further verify that the robustness of our model stems from the proposed agent architecture itself rather than from a stronger teacher model, we construct a fully LLM-free, rule-based reflection supervision signal. Specifically, for HotpotQA, we use the annotated supporting facts as evidence for the answer; for single-hop NQ, we use the final answer text as a proxy for supporting facts. When the search agent retrieves con-

Table 5: Rule-based reflection supervision signal. The validation samples are selected with the protocol of Zheng et al. (2025).

| Variants | In domain | Out of domain | AVG. |
|---|---|---|---|
| w/o reflection reward | 0.4404 | 0.4204 | 0.4261 |
| Rule-based reward | **0.4440** | 0.4376 | 0.4390 |
| Model-based reward | 0.4410 | **0.4465** | **0.4450** |

tent containing these supporting facts, we mark the retrieval as effective and set the ground-truth reflection label to True; otherwise, the label is set to False. Comparing this rule-based ground-truth reflection label with the model's own prediction yields the reflection reward. Note that the supporting facts are span-level text and may not perfectly align semantically with the retrieved passages; furthermore, the supporting-fact annotations in HotpotQA are known to be noisy. Even so, as shown in Table 5, the rule-based reward still brings a clear improvement over using no reflection reward, and its effect is comparable to that of the model-based reward. This suggests that: (i) GPT-4o-mini does not transfer high-level reasoning ability to the model here, but merely supplies a binary correctness label for the reflection; and (ii) the robustness of our method primarily arises from the RE-Searcher agent architecture and its goal-reflection mechanism, rather than the reasoning capabilities of a stronger teacher model.

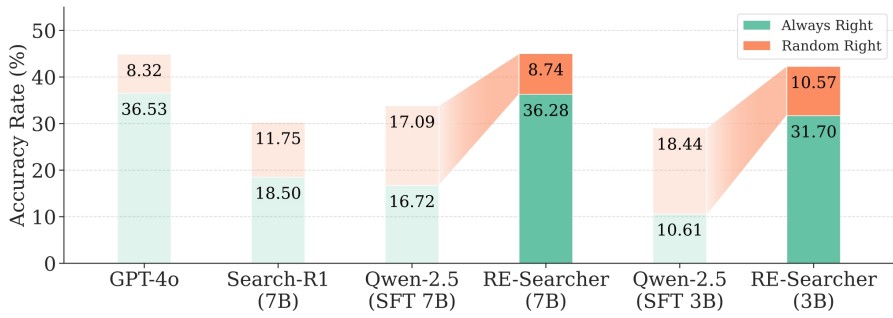

Figure 6: Analysis on the negative impacts of search fragility. The goal-reflection mechanism can effectively alleviate the negative impacts of search fragility.

Table 6: Comparison with pure planning and pure reflection. The validation samples are selected with the protocol of Zheng et al. (2025).

Table 7: Adaptive Search of RE-Searcher: uses difficulty-aware search strategies to balance efficiency and performance.

| Variants | In domain | Out of domain | AVG. |
|---|---|---|---|
| Planning only | 0.4061 | 0.4362 | 0.4339 |
| Reflection only | 0.4275 | 0.4234 | 0.4246 |
| RE-Searcher | **0.4410** | **0.4465** | **0.4450** |

| | AVG. | Std Dev | Max | Min |
|---|---|---|---|---|
| Single-hop | 1.87 | 0.94 | 5 | 1 |
| Multi-hop | 4.20 | 1.36 | 7 | 1 |
| Mixed | 3.20 | 1.08 | 7 | 1 |

### 4.2.3 FURTHER DISCUSSION ON THE GOAL-REFLECTION MECHANISM

Planning (problem decomposition) and reflection (evaluation of retrieved information) are widely used to improve model reasoning. However, when a model can interact with an external environment, it must not only reason well but also adapt to environmental dynamics. Our goal-reflection mechanism addresses this by requiring the model to set an explicit goal before each action, effectively predicting the action's outcome. After observing the actual outcome, the model compares it with its prediction, thereby learning more effective interaction with the environment.

We design experiments to show that goal-reflection is not just a simple combination of planning and reflection. We remove the goal-setting and outcome-reflection components from RE-Searcher and separately train models equipped only with planning (problem decomposition) or only with reflection (evaluation of retrieved information). As shown in Table 6, the goal-reflection model consistently outperforms these variants in both in-domain and out-of-domain settings. The results in Section 4.4 further demonstrate the superior robustness of the goal-reflection mechanism to various forms of interference.

### 4.3 NEGATIVE IMPACTS OF SEARCH FRAGILITY

We further demonstrate that the goal-reflection mechanism can effectively alleviate the negative impacts of search fragility. Fig. 6 presents the Pass@k (k=2) results for GPT-4o, Search-R1, Qwen-2.5-3B-SFT, Qwen-2.5-7B-SFT, and our RE-Searcher with both Qwen-2.5-3B-instruct and Qwen-2.5-7B-instruct as base model. In this context, the "always right" refers to the proportion of instances where all k attempts yield the correct answer, while the "random right" indicates the proportion of instances where at least one out of k attempts is correct. The results clearly showcase that through training with goal-reflection, the random right ratio is substantially reduced, particularly against Qwen-2.5-7B-SFT, where it decreased by approximately 8.35%, and even more significantly against Search-R1, with a reduction of up to 3.01%. A surprising finding is that the random right ratio of our RE-Searcher (7B) is 8.74%, remarkably close to GPT-4o's 8.32%. This proximity strongly demonstrates the effectiveness of our goal-reflection mechanism in alleviating search fragility.

### 4.4 ROBUSTNESS AGAINST EXTERNAL DISTURBANCES

We demonstrate that our proposed framework significantly enhances the model's robustness against external disturbances. To simulate real-world noise, we intentionally introduce disturbances to the queries during the first round of the search process. This is designed to both misdirect the

initial search direction and challenge the model's corrective capabilities. Specifically, we randomly employ one of the following three types of disturbances: i) Randomly reducing a word: A word is randomly removed from the query. ii) Randomly adding a word: A random word is inserted into the query. iii) Randomly replacing a word with similar semantics: A word is replaced by another with a similar meaning. All these disturbance operations are implemented by prompting GPT-4o-mini. We then compare the proportion of instances that transition from correct to incorrect after noise injection, effectively measuring the degradation caused by disturbances. The results, presented in Fig. 7,

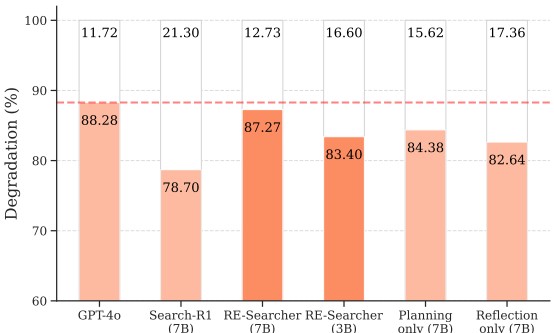

Figure 7: Robustness analysis against disturbances. Our RE-Searcher exhibits a lower degradation.

show that our RE-Searcher exhibits a substantially lower degradation compared to Search-R1. Specifically, our framework achieves an improvement of -8.57% in degradation relative to the Search-R1 with the same size base model. Furthermore, even our 3B model outperforms the Search-R1 (7B) in terms of robustness. Notably, our RE-Searcher (7B) achieves a comparable degradation to GPT-4o, further underscoring the superior ability of our goal-reflection mechanism to improve robustness against external disturbances. In addition, the robustness of the 7B models trained with pure planning and pure reflection mechanisms is substantially lower than that of RE-Searcher (7B), and is only comparable to RE-Searcher (3B). This further underscores the performance gap between the goal-reflection mechanism and simple task decomposition or self-reflection strategies.

### 4.5 ADAPTIVE SEARCH STRATEGY

Training models with enhanced reasoning capabilities often introduces an unintended side effect: when presented with simple problems, the model may engage in unnecessary overthinking, leading to high computational cost and increased inference latency (Chen et al., 2024; Sui et al., 2025). Consequently, adaptive reasoning has become a central focus in recent research. We analyze the number of search rounds performed by RE-Searcher (7B) across different types of data. As shown in Table 7, the number of searches varies substantially across datasets of different difficulty levels. The goal-reflection mechanism enables the model to terminate the search process once it has obtained results that are sufficiently relevant to the goal, thereby allowing it to conclude early and avoid superfluous computation.

## 5 DISCUSSION AND CONCLUSION

In this paper, we investigate the instability of search agents during search and problem-solving. We identify a critical issue: complex external environments can amplify small initial errors into large deviations in the final output. To address this, we propose RE-Searcher, a novel search agent that integrates goal setting with outcome reflection to counteract the fragility of search processes in complex environments. Through extensive numerical and perturbation experiments, we demonstrate that our approach substantially improves the robustness of search agents. Nevertheless, we acknowledge that this work represents an initial step. The proposed training methodology is relatively simple, and there is considerable scope for enhancement. Future improvements could involve refining the training data, advancing the learning algorithms, and designing more sophisticated supervision signals. We believe that with these enhancements, the agent's performance in complex environments can be further elevated.

Looking ahead, the rapid progress of LLM-powered agents is enabling them to operate across an ever-wider array of external environments, i.e., often more complex and dynamic than before. While we embrace the convenience and capabilities that greater agent autonomy brings, we must also pay close attention to the complex and potentially unintended consequences of their interactions with the environment. Our future work will delve deeper into these potential issues, aiming to foster the sustainable and responsible advancement of autonomous agents.

ETHICS STATEMENT

This study does not involve human sujects, sensitive information, or any applications with foreseeable ethical issues. We have thoroughly reviewed the ICLR Code of Ethics and affirm that our work fully complies with its requirements.

REPRODUCIBILITY STATEMENT

To ensure reproducibility, we provide comprehensive descriptions of all methods in the main text. The experimental section details the computational environment, datasets, algorithms, and all hyperparameter settings. In addition, we include our code and accompanying documentation in the supplementary materials.

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

# A  APPENDIX

## A.1  RELATED WORKS

Integrating external data is a pivotal strategy for overcoming the inherent limitations of Large Language Models (LLMs), notably knowledge cutoff and hallucination. The prevailing approaches can be broadly categorized into two paradigms: passive Retrieval-Augmented Generation (RAG) and proactive agentic search.

### A.1.1  RETRIEVAL-AUGMENTED GENERATION

Traditional RAG frameworks enhance model outputs by retrieving relevant information from an external corpus. This is typically achieved by encoding queries and knowledge passages into a shared vector space and fetching the nearest neighbors to augment the generation process for complex tasks (Ma et al., 2023; Arslan et al., 2024). A significant drawback of these methods is their reliance on static, manually engineered prompts and workflows. Recent efforts have sought to improve RAG along two primary axes. On the retrieval front, methodologies like LightRAG (Guo et al., 2024) and GraphRAG (Edge et al., 2024) leverage knowledge graphs to structure external data, facilitating more precise and contextually relevant information retrieval. On the generation front, works such as IRCoT (Trivedi et al., 2022a) integrate Chain-of-Thought (CoT) reasoning to refine both information seeking and synthesis. Meanwhile, AirRAG (Feng et al., 2025) employs Monte Carlo Tree Search (MCTS) to systematically explore diverse information pathways. Despite these advancements, these models remain fundamentally reactive; they do not proactively strategize on query formulation or dynamically adapt their reasoning based on retrieved results.

### A.1.2  AGENTIC SEARCH-AUGMENTED MODELS

A recent surge of interest has focused on developing autonomous agents that treat search engines as callable tools to support sophisticated reasoning. This agentic search paradigm for question-answering (QA) places a high demand on a model's planning and reasoning faculties, leading many researchers to turn to reinforcement learning (RL) for training. For instance, a series of works including Search-R1 (Jin et al., 2025), DeepResearcher (Zheng et al., 2025), and R1-Searcher++ (Song et al., 2025) have successfully applied RL algorithms like GRPO to train agents for multi-hop QA (Yang et al., 2018; Kwiatkowski et al., 2019), significantly boosting their search and inference performance. StepSearch (Wang et al., 2025b) refines this approach by introducing step-wise reward signals within a PPO framework, incentivizing productive actions at each stage of the search. Concurrently, DynaSearcher (Hao et al., 2025) pioneers a dynamic knowledge graph that evolves during the search to guide exploration, while also leveraging heterogeneous data sources to enrich the agent's knowledge base. These contributions have substantially propelled the field forward, enabling models to more adeptly harness external knowledge for reasoning.

In this work, we build upon these foundations by performing a rigorous analysis of the search fragility brought by the complex search environment. We introduce a novel search agent designed to foster greater robustness during information retrieval, thereby elevating the quality and reliability of the model's final responses.

## A.2  EXPERIMENTS DETAILS FOR PRELIMINARY ANALYSIS

### A.2.1  STOCHASTICITY ANALYSIS

To investigate the instability of search agents during the search process, we constructed agents based on three distinct models: GPT-4o, Qwen2.5 3B, and Qwen2.5 7B. To ensure that the Qwen2.5 models produced outputs in the required format, we fine-tuned them using the warm-up data detailed in Section 3.1. Our evaluation was conducted on a dataset of 3,197 instances selected by Zheng et al. (2025), with Exact Match (EM) serving as the primary metric for accuracy. Each agent was run $k = 2$ times on the dataset. We categorize the outcomes as follows: questions answered correctly in all trials are labeled "always right," while those answered correctly in some but not all trials are labeled "random right." We then calculated the proportions of "always right" ($P_{AR}$) and "random right"($P_{RR}$) questions by dividing their respective counts by the total number of questions:

$$P_{\text{AR}} = \frac{1}{N} \sum_{i=1}^{N} \mathbf{1}\Big\{ \sum_{r=1}^{k} c_i^{(r)} = k \Big\} \tag{4}$$

$$P_{\text{RR}} = \frac{1}{N} \sum_{i=1}^{N} \mathbf{1}\Big\{ 1 \leq \sum_{r=1}^{k} c_i^{(r)} \leq k-1 \Big\} \tag{5}$$

Where N is the total number of the instances. $c_i^{(r)}$ is an indicator variable representing whether the answer is correct for sample $i$ in trial $r$, where a correct answer is recorded as 1 and an incorrect answer is recorded as 0.

### A.2.2 FRAGILITY ANALYSIS

To quantify the impact of minor variations in search queries on the search results, we introduce three types of single-word perturbations to the keywords within the model's search trajectory: **synonym substitution**, **keyword addition**, and **keyword deletion**. We use the search engine from Jin et al. (2025) to retrieve results for both the original and the perturbed queries, yielding search result $R$ and $R'$, respectively. Subsequently, we employ the all-MiniLM-L6-v2 model[4] to encode each set of search results into a dense vector representation. The similarity between the original and perturbed results is then measured by computing the cosine similarity of their corresponding vectors. The formula for calculating this search result similarity is as follows:

$$S(R, R') = \cos(\theta) = \frac{\vec{v} \cdot \vec{v}'}{\|\vec{v}\|\|\vec{v}'\|} \tag{6}$$

where $S(R, R')$ represents the final similarity score between the original search results $R$ and the perturbed search results $R'$. $\vec{v}$ and $\vec{v}'$ represents the vector embedding of the original search results $R$ and perturbed search results $R'$ respectively.

### A.2.3 DEEPER ANALYSIS OF SEARCH FRAGILITY

Under the setting in Section 4.3, using only two samples per question may yield a biased view. We therefore increase the number of samples to four per question under the same configuration. As shown in Fig. 8, RE-Searcher stays very close to GPT-4o in both the 2-sample and 4-sample settings. With only 7B parameters, it matches the performance of the large proprietary GPT-4o model, indicating that our conclusions are not an artifact of using only two samples.

Search-R1 clearly benefits from more samples: its random-right score improves markedly from 2 to 4 samples and becomes close to that of RE-Searcher and GPT-4o, showing that more samples can partially mitigate single-shot instability. However, its upper bound 32.90% (always right + random right) remains noticeably lower than that of GPT-4o and RE-Searcher, leading to a worse random-right rate 50% (random right / union right). This suggests that Search-R1 is limited both in "always right" behavior and in the additional correct behavior recoverable through sampling.

By contrast, the Qwen2.5 models adopt a much more aggressive strategy with 4 samples: they improve their up bound mainly by sharply increasing random right, while their always-right scores drop substantially. This strong trade-off between stability and upper-bound performance is exactly the "fragility" we study: the agents' behaviors and answers vary greatly across samples.

Overall, increasing the number of samples from 2 to 4 does not remove the inherent fragility discussed in Section 2.1 and Section 4.3. Models still show substantial across-sample variability, closely tied to their strategy (conservative vs. aggressive) and overall performance.

### A.2.4 NECESSITY OF WARM-UP

---

[4]https://huggingface.co/sentence-transformers/all-MiniLM-L6-v2

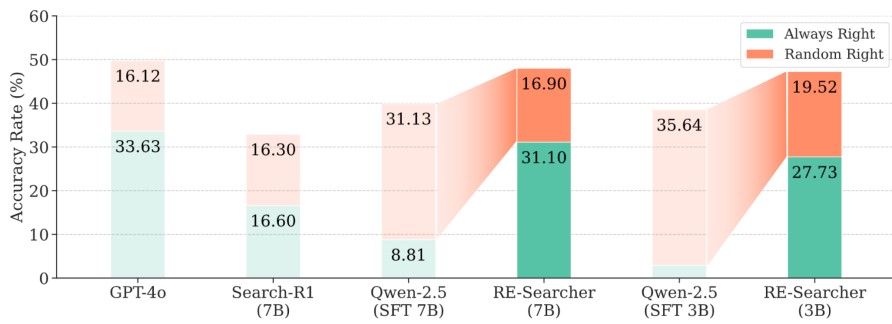

Figure 8: Further Analysis on the negative impacts of search fragility: each question was sampled 4 times.

To verify whether models like Qwen2.5 (3B) can follow our relatively simple instruction template without additional warm-up, we conducted an ablation experiment in which the supervised warm-up phase was removed and the model was trained directly using reinforcement learning ("no warm-up").

In the no-warm-up setting, the format reward remained consistently low across RL training, indicating that the model failed to reliably follow the predefined response template. As shown in Fig. 9, the format reward of the no-warm-up model plateaus around 0.05–0.10, whereas the model trained with the warm-up stage reaches around 0.30–0.50

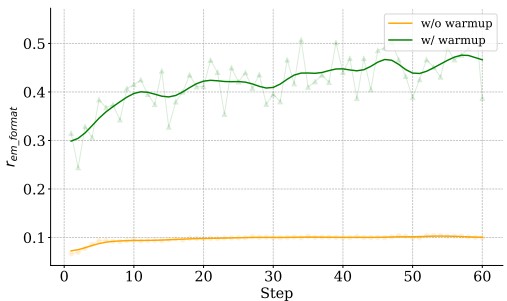

Figure 9: Format reward of Qwen2.5 (3B) during the RL training process.

These observations suggest that, in our RL setup, the model's generic instruction-following ability is not sufficient to guarantee stable learning of the specific formatting constraints. The warm-up stage provides the model with explicit supervised signals on the target template, which (i) significantly improves the format reward and (ii) leads to more stable and reliable RL training. Therefore, the warm-up is empirically necessary in our framework, even for a relatively capable base model like Qwen2.5-3B.

### A.2.5 SENSITIVITY ANALYSIS OF REFLECTION REWARD COEFFICIENT

In Eq. (3), we empirically set the coefficient of the reflection reward to 0.1 in order to balance the contributions of the different reward components. To further validate this choice and obtain better training hyperparameters, we conduct a sensitivity analysis with respect to this coefficient. The results, summarized in Table 8, show that using 0.1 as the coefficient yields the best overall performance in terms of average score, and in particular leads to the largest gains on out-of-domain data. Moreover, all non-zero coefficients (0.1,

Table 8: Sensitivity analysis of coefficient of reflection reward. The validation samples are selected with the protocol of Zheng et al. (2025).

| coefficient | In domain | Out of domain | AVG. |
|---|---|---|---|
| 0 | 0.4404 | 0.4204 | 0.4261 |
| 0.1 | 0.4415 | **0.4465** | **0.4451** |
| 0.3 | 0.4360 | 0.4248 | 0.4280 |
| 0.5 | **0.4425** | 0.4398 | 0.4406 |

0.3, 0.5) outperform the setting without the reflection reward (0.0), suggesting that our approach is not overly sensitive to the precise value of this weight and that the reflection reward consistently provides performance benefits.

### A.2.6 STATISTICS OF OUTPUT TOKENS AND SEARCH STEPS

Table 9: Statistics of output tokens.

| Model | AVG. | Std Dev | Max | Min |
|---|---|---|---|---|
| Search-R1-instruct (7B) | 186.23 | 156.33 | 1331 | 48 |
| RE-Searcher (7B) | 330.46 | 111.47 | 906 | 73 |

Table 10: Statistics of search steps.

| Model | AVG. | Std Dev | Max | Min |
|---|---|---|---|---|
| Search-R1-instruct (7B) | 1.72 | 1.31 | 5 | 1 |
| RE-Searcher (7B) | 3.20 | 1.08 | 7 | 1 |

We report the number of output tokens and search steps for RE-Searcher and Search-R1. Our method indeed leads to a higher average number of generated tokens and search steps. However, an average of around 330 output tokens and about 3 search steps per question remains practically acceptable. Moreover, our method improves the stability of the model (with lower Std Dev), making its behavior more reliable on some particularly challenging queries.

### A.3 USE OF LARGE LANGUAGE MODELS

We used a large language model solely for copyediting purposes, i.e., correcting typographical errors, refining grammar, and polishing the prose. No other aspects of this work employed LLMs.

