# OpenReview forum: "RE-Searcher: Robust Agentic Search via Goal-oriented Planning and Self-reflection"
_ICLR.cc/2026/Conference — Submitted to ICLR 2026_

### Official Review · Reviewer_i3qY · 2025-10-29

**Soundness:** 2
**Presentation:** 3
**Contribution:** 3
**Rating:** 4
**Confidence:** 5

**Summary:**

1. This paper focuses on agentic search, analyzing the complexity and variability of search environments as well as the fragility of search agents. It argues that search agents should incorporate explicit goal clarification and reflection on search results like humans.

2. This work introduces RE-Searcher, which integrates explicit goal clarification and reflection on search results into the agent loop. It further designs a reflection reward, employing a stronger LLM-as-Judge to assess reflection quality, and trains the search agents using the GRPO algorithm.

3. Extensive experiments across in-domain and out-of-domain QA benchmarks show that RE-Searcher consistently outperforms recent agentic search baselines (e.g., Search-R1, ZeroSearch, O2-Searcher), with notable robustness under perturbations.

**Strengths:**

1. Good motivation and preliminary analysis: The paper clearly identifies the challenges of environmental variability and inherent fragility faced by search agents, and conducts preliminary experiments to validate these issues.

2. Clear methodological design: In response to the identified problems, the paper extends the agent loop and proposes an effective training strategy. The methodological descriptions and figures are well-presented and easy to follow.

3. Strong results: The experiments compare against a wide range of baselines and achieve the best overall performance. Extensive ablation and robustness analyses further demonstrate how effectively the proposed method addresses the original problem.

**Weaknesses:**

1. In Sections 2.1 and 4.4, the paper examines the inherent fragility of search agents. While I acknowledge the stochasticity in search agent outputs, sampling only twice per question may not be sufficient to support such a conclusion.

2. As I understand it, there may be an issue with Equation (2). The function FM is not explicitly defined, and I assume it is a 0–1 function similar to EM. When EM equals 1, if the format is correct (FM=1), the reward r_{em_format}=1-0.2\*1=0.8. However, if the format is incorrect (FM=0), the reward becomes r_{em_format}=1-0.2\*0=1, which paradoxically yields a higher reward for incorrect formatting.

3. I think using LLM-as-Judge to provide reward supervision for the newly introduced reflection action is a very good choice. However, I noticed in Equation (3) that the reflection reward is added to the outcome reward. This makes me wonder:  since reflection is a process-level action, why not treat it as a process reward instead? Theoretically, process rewards are more fine-grained and could improve sample efficiency. In addition, how was the weight of 0.1 in Equation (3) determined?

4. Based on your main experimental results (Table 2), since all datasets are open-domain QA, why does the method achieve substantial improvements on some datasets but not on others? Have you analyzed the reasons behind this phenomenon?

**Questions:**

Please refer to Weaknesses. I am open to changing the score based on your rebuttal.

---

> ### Author Response · Authors · 2025-11-21
>
> # Experiments of 4 sampling (Weakness 1)
>
> We thank the reviewer for raising this concern. We agree that using only 2 samples per question may be insufficient to fully support claims about the inherent fragility of search agents. To address this, we ran additional experiments with 4 samples per question, using a temperature of 0.6 for all models. The results are shown below. Here, 1) @2 denotes 2 samples per question; 2) @4 denotes 4 samples per question; 3) union right = always right + random right; 4) random rate = random right/union right.
>
> | Model         | always right@2 ↑ | always right@4 ↑ | random right@2 ↓ | random right@4 ↓ | union right@2 ↑ | union right@4 ↑ | random rate@2 ↓ | random rate@4 ↓ |
> |---------------|----------------|----------------|----------------|----------------|---------------|---------------|---------------|---------------|
> | GPT-4o        | 36.53          | 33.63          | 8.32           | 16.12          | 44.85         | 48.65         | 0.19          | 0.33          |
> | Search-R1-7B  | 18.50          | 16.60          | 11.75          | 16.30          | 30.25         | 32.90         | 0.39          | 0.50          |
> | Qwen2.5-SFT-7B| 16.72          | 8.81           | 17.09          | 31.13          | 33.81         | 39.94         | 0.51          | 0.78          |
> | RE-Searcher-7B| 36.28          | 31.10          | 8.74           | 16.90          | 45.02         | 48.00         | 0.19          | 0.35          |
> | Qwen2.5-SFT-3B| 10.61          | 2.90           | 18.44          | 35.64          | 29.05         | 38.54         | 0.63          | 0.92          |
> | RE-Searcher-3B| 31.70          | 27.73          | 10.57          | 19.52          | 42.27         | 47.25         | 0.25          | 0.41          |
>
> Our key observations are:
> 1. RE-Searcher remains very close to GPT-4o under both the 2-sample and 4-sample settings. With only 7B parameters, it achieves performance comparable to the large proprietary GPT-4o model. This indicates that our conclusions are not an artifact of using only 2 samples.
> 2. Search-R1 benefits from more samples: its random right improves substantially when moving from 2 to 4 samples and becomes close to RE-Searcher and GPT-4o. This confirms that increasing the number of samples can partially mitigate single-shot instability. However, its performance upper bound, reflected by union right, is still clearly lower than that of GPT-4o and RE-Searcher, leading to a noticeably worse random rate. This suggests that Search-R1 is limited both in terms of “always right” behavior and the additional correct behavior recoverable via sampling.
> 3. The Qwen2.5 models adopt a much more aggressive strategy with 4 samples: they achieve higher union right mainly by sharply increasing random right, while their always right scores drop considerably. This strong trade-off between stability and upper-bound performance is precisely a manifestation of the “fragility” we study: across different samples, the agents’ behaviors and answers vary substantially.
>
> Overall, increasing the number of samples from 2 to 4 does not remove the inherent fragility discussed in Sections 2.1 and 4.4. Models still exhibit significant across-sample variability, and this variability is closely tied to their strategy (conservative vs. aggressive) and overall performance. We will incorporate these 4-sample results and the corresponding analysis into Sections 2.1 and 4.4 of the final version to more thoroughly support our conclusions.
>
> # Issue with Equation (2) (Weakness 2)
> Thank you very much for your careful reading and for pointing out this issue.
> $FM$ is indeed intended to be a 0–1 indicator function that denotes whether the output format is correct. You are absolutely right that, in its current form, Equation (2) is problematic. The term $1 − 0.2 * FM$ is a typo and does not reflect our original design.
> Our original intention is that, when $EM = 1$, an incorrect format ($FM = 0$) should be penalized by 0.2. Therefore, the correct expression should be:
> $$1 − 0.2 * (1 − FM) = 0.8 + 0.2 * FM$$
> In the revised version, we will:
> 1. Explicitly define $FM$ as a 0–1 indicator function; and
> 2. Correct Equation (2) by replacing $1 − 0.2 * FM$ with $1 − 0.2 * (1 − FM)$ (equivalently, $0.8 + 0.2 * FM$), so that it is consistent with our implementation and intended behavior.
>
> We sincerely appreciate your comment. It helps us improve both the clarity and the correctness of the paper.

---

> ### Author Response · Authors · 2025-11-21
>
> # Application of process-level rewards (Weakness 3)
>
> Thank you for this very insightful comment. In our framework, the reflection reward is indeed a process-level reward. However,  the training algorithm, GRPO, estimates the advantage function through group-wise regularization to reduce variance during training. This structure makes it impractical to apply group-wise regularization at every individual step, and thus, we choose to place supervised signals in the outcome reward. Concretely, the outcome reward for the whole trajectory is used to compute a single episode-level advantage value, and this single advantage is then broadcast to all action tokens in the trajectory during optimization. For this reason, in Eq. (3), we aggregate the reflection reward into the outcome reward: although the reflection reward is defined at the process level, it still influences the optimization of the entire trajectory through this shared episode-level advantage in GRPO.
>
> We additionally conducted an experiment that attempts to incorporate supervision more explicitly into the training process by forcing process rewards into GRPO without considering the increased variance. Concretely, on top of the standard episode-level advantage, we applied an additional scaling to the advantages of the reflection tokens only, in the form:
>
> $$A_{reflect} := (1 + β · r) · A$$
>
> where $r$  is the reflection reward and $A$ is the original advantage. Intuitively, this is meant to upweight the contribution of reflection tokens in the gradient updates. However, this modification ignores the variance-control design of GRPO, is largely heuristic, and lacks solid theoretical support. In practice, this setup collapsed after about 20 training steps, making RL training highly unstable.
>
> As for the choice of the reward coefficients, we initially set the reflection reward weight to 0.1 based on preliminary experiments and domain experience, aiming to balance the reflection reward against the other reward components. To address the reviewer’s concern more systematically, we have now conducted a sensitivity analysis over different coefficients. The results are summarized below:
>
> | coefficient | In domain | Out of domain | AVG.  |
> |------------|-----------|---------------|-------|
> | 0          | 0.4404    | 0.4204        | 0.4261 |
> | 0.1        | 0.4415    | **0.4465**    | **0.4451** |
> | 0.3        | 0.4360    | 0.4248        | 0.4280 |
> | 0.5        | **0.4425**| 0.4398        | 0.4406 |
>
> The results show that using 0.1 achieves the best overall performance in terms of the average score, and especially improves out-of-domain performance. In addition, all non-zero coefficients (0.1, 0.3, 0.5) outperform the setting without a reflection reward (0.0), which indicates that our method is not overly sensitive to the exact value of this weight and that the reflection reward consistently brings benefits. We will include this sensitivity study in the final version of the paper.

---

> ### Author Response · Authors · 2025-11-21
>
> # Heterogeneous improvements on different datasets (Weakness 4)
>
> We appreciate the reviewer’s careful observation. The heterogeneous gains across datasets are mainly due to the different reasoning requirements of each benchmark.
>
> First, datasets such as NQ, TriviaQA, and PopQA are predominantly **single-hop QA tasks**. Correct answers can often be obtained by **retrieving one relevant passage** (The search keywords may be contained within the question.) and performing shallow reasoning, so **they put relatively limited emphasis on complex reasoning and reflection**. Since RE-Searcher is specifically designed to improve iterative reasoning and reflection over multiple search steps, its advantage is naturally less pronounced on these simpler single-hop datasets than on multi-hop benchmarks such as HotpotQA, 2Wiki, Musique, and Bamboogle.
>
> Second, even on these single-hop datasets, our method performs competitively: RE-Searcher is slightly worse than ZeroSearch but remains comparable to or better than most other baselines (see Table 2). **The gap to ZeroSearch is due to its use of LLM as a simulated search engine**. ZeroSearch finetuned an LLM to directly predict search queries and their corresponding passages, and then uses this simulation LLM to replace the external search engine. Compared with a standard retrieval engine, such a simulation LLM has much stronger semantic matching ability and can often “retrieve” highly relevant information in one step. This design gives ZeroSearch a particular advantage on single-hop QA, where semantic matching is usually sufficient.
>
> However, on multi-hop datasets, strong semantic matching alone is not enough. The key challenge is to decompose the question into intermediate sub-queries and iteratively update them based on previous evidence. In this setting, our explicit reasoning–reflection loop is crucial for forming better search queries and using retrieved information more effectively. Consequently, RE-Searcher consistently outperforms all baselines, including ZeroSearch, on multi-hop benchmarks, leading to the best average performance across all datasets.
>
> Thank you very much for these constructive suggestions, which have significantly enriched the content and perspective of our paper. Below, we summarize several key comments from other reviewers and our corresponding responses. These points were also highly beneficial to our work and may be of interest to you:
>
> - **Dependency on external LLMs (Reviewer 4Xdc)**: Although we use GPT-4o-mini to supervise the reflection signal, this supervision is only applied to the final reflection output of the model (i.e., the <reflect>True/False</reflect> tokens). It only judges whether the reflection is correct, and does not involve any leakage of external knowledge. We further added a rule-based reward experiment, which demonstrates that the performance gains of RE-Searcher mainly come from methodological improvements rather than knowledge leakage from a more powerful model.
> - **Computational cost of agent inference (Reviewers 4Xdc and tB4e)**: We report the number of output tokens and search steps for RE-Searcher and Search-R1. Our method indeed leads to a higher average number of generated tokens and search steps. However, an average of around 330 output tokens and about 3 search steps per query remains practically acceptable. Moreover, our method improves the stability of the model, making its behavior more reliable on some particularly challenging queries.
> - **Deeper explanation of the novelty (Reviewer tB4e)**: While planning (decomposing and organizing tasks) and reflection are common techniques in agent systems, our proposed goal planning is not a traditional task decomposition. Instead, it sets an explicit goal for the current action, akin to predicting the outcome of that action. Goal planning and reflection together form a prediction–action–reflection loop, enabling the model to interact and learn from the external environment in a manner similar to a world model, thus enhancing its robustness to environmental variability. We have added experiments comparing our approach with pure planning and pure reflection baselines, and the results confirm the advantages of our method.
>
> Once again, we sincerely thank all reviewers for their insightful comments. They have provided us with a more comprehensive view of our work and helped us refine and deepen our study. We will incorporate these new analyses and results into the final version of our manuscript.

---

> ### Author Response · Authors · 2025-11-27
>
> Dear Reviewer i3qY,
>
> We sincerely appreciate your valuable feedback on the paper. Given that the deadline for the discussion phase is approaching, we would be most grateful if you could kindly share any remaining concerns at your earliest convenience, and we will address them immediately.

---

### Official Review · Reviewer_tB4e · 2025-11-01

**Soundness:** 3
**Presentation:** 4
**Contribution:** 2
**Rating:** 6
**Confidence:** 3

**Summary:**

This paper addresses the fragility of LLM-based search agents in complex environments where small query variations can lead to drastically different retrieval results and erroneous reasoning trajectories. The authors propose RE-Searcher, an agentic framework designed to improve robustness by integrating goal-oriented planning and self-reflection. During its search process, the agent explicitly articulates a search goal, retrieves information, and then reflects on whether the retrieved evidence successfully satisfies that goal. The agent is trained using GRPO with a composite reward signal that includes factual correctness, format adherence, and a reflection accuracy score provided by an "LLM as Judge". Experimental results show that RE-Searcher achieves state-of-the-art performance on several question-answering benchmarks and demonstrates significantly improved robustness against search fragility and external query perturbations.

**Strengths:**

- The motivation is clear. The preliminary analysis in Section 2.1 quantifies output stochasticity across different model scales. This empirically demonstrates a critical instability problem that fundamentally limits achievable performance.
- The proposed method is intuitive and clearly described in Section 3. The structured generation template clearly defines three discrete actions.
- Robustness Analysis is novel. The Pass@2 analysis in Section 4.3 demonstrates that RE-Searcher substantially reduces the random-right ratio.

**Weaknesses:**

- The novelty of core concepts is limited. The core ideas of "planning" (decomposing a problem) and "reflection" (evaluating retrieved information) are well-established concepts in agentic AI and RAG literature.

- The proposed method introduces multiple new components for each search iteration: an explicit "think" step, a "goal" generation step, and a "reflect" step. This multi-step process seems guaranteed to significantly increase the number of generated tokens and overall inference latency compared to simpler RAG or SFT baselines.

- While the paper ablates the reflection reward, there is no ablation isolating the contribution of goal-oriented planning versus self-reflection. The method combines both mechanisms, but their individual contributions remain unclear.

**Questions:**

See weaknesses above.

---

> ### Author Response · Authors · 2025-11-21
>
> # Novelty of our proposed method (Weakness 1 & Weakness 3)
> We thank the reviewer for the insightful comment. We agree that "planning" and "reflection" are well-established concepts in the agentic AI literature. However, we would like to clarify that the novelty of our approach lies in the specific instantiation and synergistic interplay of these concepts to address the unique challenges of search agents.
>
> 1. **The Role of "Goal-Planning" is Distinct from Conventional Task Decomposition**. In conventional agent systems, "planning" typically serves to decompose a complex task into smaller, manageable sub-tasks. In contrast, our "goal-planning" module is designed to set a specific search goal for each query. This is more akin to a prediction of the expected outcome of a search action. For instance, the agent predicts, "by performing this search query, I expect to find information about X." If the actual search result deviates from this predicted goal, the reflection mechanism is triggered to analyze the discrepancy and refine the subsequent actions. This predict-act-reflect paradigm, which is prevalent in the world model research, enables the agent to build a more accurate understanding of the complex and dynamic external information environment.
> 2. **The Inherent Synergy between Goal-Planning and Reflection**. As we discussed in Section 2, the volatile nature of the external environment often makes an agent's search process fragile. To achieve robust search performance, the agent must develop a deeper understanding of this environment. Our integrated goal-planning and reflection mechanism is specifically designed for this purpose. They are not two separate, parallel components but are intrinsically linked: the goal set by planning serves as the direct benchmark for reflection.
> 3. **Ablation Studies Demonstrate the Effectiveness of Our Integrated Approach**. To further validate our design, we conducted additional ablation studies to compare our full model with "pure planning" and "pure reflection" variants.
>       - Pure Planning: We have removed the original goal-setting and reflection mechanismss in our framework, instead, adopted the planning and decomposing mechanism as you requested.
>       - Pure Reflection: We trained an agent to reflect on the search results without an explicit, pre-defined goal.
>
> The results are presented in the table below.
>
> |                      | In domain | Out of domain | AVG.  |
> |----------------------|----------:|-------------:|------:|
> | Pure planning        | 0.4061    | 0.4362       | 0.4339 |
> | Pure reflection      | 0.4275    | 0.4234       | 0.4246 |
> | RE-Searcher   | **0.4410**| **0.4465**   | **0.4450** |
>
> As shown, our integrated method consistently outperforms both baselines, demonstrating that the synergy between our specific goal-planning and reflection is crucial for achieving superior performance.
>
> In addition, we conducted disturbances experiments on pure planning and pure reflection model, referring to the settings in Sec. 4.4. The experimental results are shown in the table below. Indeed, the anti-disturbances capabilities of pure planning and pure reflection are lower than those of RE-Searcher.
>
> | Model            | Accuracy Rate (%) | Degration (%) |
> |------------------|-------------------|---------------|
> | GTP-4o           | 88.28             | 11.72         |
> | Search-R1 (7B)   | 78.70             | 21.30         |
> | RE-Searcher (7B) | 87.27             | 12.73         |
> | RE-Searcher (3B) | 83.40             | 16.60         |
> | Pure Planning    | 84.38             | 15.62         |
> | Pure Refleciton  | 82.64             | 17.36         |
>
> In summary, while the conceptual building blocks of our method may seem familiar, our contribution lies in diagnosing a critical challenge for search agents and proposing a simple yet effective mechanism that repurposes these concepts into a synergistic "predict-act-reflect" loop. We believe this work offers significant insights into improving the stability and robustness of search agents.

---

> ### Author Response · Authors · 2025-11-21
>
> # Inference latency of RE-Searcher (Weakness 2)
>
> We thank the reviewer for raising the concern about increased latency. We agree that explicit goal planning and reflection introduce additional tokens. The output token statistics are:
>
> | Model         | AVG.  | Std Dev | Max | Min |
> |---------------|-------|--------|-----|-----|
> | RE-Searcher-7B | 330.46 | 111.47 | 906 | 73  |
> | Search-R1-7B   | 186.23 | 156.33 | 1331 | 48 |
>
> RE-Searcher indeed has a higher average output length, but ≈330 tokens remain practical for agent-style search. At the same time, it has a smaller variance and a lower maximum length, indicating more stable and predictable behavior. In difficult cases, the goal-planning and reflection steps help the model converge more quickly instead of repeatedly making unsuccessful attempts, mitigating extremely long trajectories.
>
> We would also like to clarify the difference between our setting and standard RAG, which makes a direct latency comparison to “simpler RAG” baselines inappropriate.
> - In RAG, the system typically performs a single retrieval from a vector database using the user query and then asks the model to summarize/organize the retrieved passages. This mainly tests information aggregation when the answer is already closely aligned with the query.
> - Our method is a **multi-hop search agent**: the initial question is often not directly aligned with the final answer, and the agent must iteratively 1) set intermediate goals, 2) issue multiple search queries, and 3) reason over intermediate results to decide the next step.  This primarily tests long-horizon reasoning and planning rather than single-step retrieval.
>
> This difference is reflected in our results (Table 2): RAG-based methods perform well on single-hop datasets such as NQ, PopQA, and TriviaQA, but degrade substantially on multi-hop datasets, where our agent maintains strong performance.
>
> In summary, while our approach modestly increases average token usage, it improves stability, avoids pathological long runs, and is designed for more challenging multi-hop search scenarios where simple RAG or SFT methods are inadequate.

---

> ### Author Response · Authors · 2025-11-21
>
> Thank you very much for these constructive suggestions, which have significantly enriched the content and perspective of our paper. Below, we summarize several key comments from other reviewers and our corresponding responses. These points were also highly beneficial to our work and may be of interest to you:
>
> - **Dependency on external LLMs (Reviewer 4Xdc)**: Although we use GPT-4o-mini to supervise the reflection signal, this supervision is only applied to the final reflection output of the model (i.e., the <reflect>True/False</reflect> tokens). It only judges whether the reflection is correct, and does not involve any leakage of external knowledge. We further added a rule-based reward experiment, which demonstrates that the performance gains of RE-Searcher mainly come from methodological improvements rather than knowledge leakage from a more powerful model.
> - **Heterogeneous improvements on different datasets (Reviewers 4Xdc and i3qY)**: In Table 2, the performance gains of our model vary across datasets. This is mainly because NQ, TriviaQA, and PopQA are single-hop datasets: one search is usually sufficient to obtain relevant information, so the demand for sophisticated reflection behavior is relatively low. Consequently, our improvements on these datasets are less pronounced than on multi-hop datasets. In contrast, ZeroSearch employs a model-based search engine with stronger semantic matching capabilities and therefore performs better on single-hop data. Nonetheless, our approach still achieves competitive performance on single-hop datasets and attains SOTA results on average across all benchmarks.
> - **Sensitivity analysis of the reward coefficient (Reviewers 4Xdc, SaqS, and i3qY)**: We initially set the reflection reward coefficient to 0.1 based on empirical considerations to balance the contributions of different reward components. We have now added a sensitivity analysis experiment on this coefficient, and the results support our original choice.
> - **Application of process-level rewards (Reviewer i3qY)**: In our framework, the reflection reward is indeed defined at the process level, but the GRPO algorithm estimates a single episode-level advantage via group-wise regularization to reduce variance. This makes it impractical to incorporate step-wise supervision directly into GRPO, so we aggregate the reflection reward into the outcome reward in Eq. (3), and the resulting episode-level advantage is then broadcast to all action tokens in the trajectory. We also tried a more explicit but heuristic scheme that selectively upweighted the advantages of reflection tokens, i.e., $A_{reflect} := (1 + β · r) · A$. However, this breaks the variance-control design of GRPO and leads to instability, with RL training collapsing after about 20 steps. Based on these observations, we adopted the current design (adding reflection rewards to the outcome reward), which is significantly more stable and achieves better performance in our experiments.
>
> Once again, we sincerely thank all reviewers for their insightful comments. They have provided us with a more comprehensive view of our work and helped us refine and deepen our study. We will incorporate these new analyses and results into the final version of our manuscript.

---

> ### Author Response · Authors · 2025-11-27
>
> Dear Reviewer tB4e,
>
> We sincerely appreciate your valuable feedback on the paper. Given that the deadline for the discussion phase is approaching, we would be most grateful if you could kindly share any remaining concerns at your earliest convenience, and we will address them immediately.

---

### Official Review · Reviewer_SaqS · 2025-11-01

**Soundness:** 2
**Presentation:** 3
**Contribution:** 2
**Rating:** 4
**Confidence:** 4

**Summary:**

This paper addresses a critical challenge in LLM augmented with search tools: agent LLMs tend to generate variable and inconsistent results.  Authors first conducts a systematic analysis showing that semantically plausible query variations can lead to dramatically divergent and unreliable search trajectories. To mitigate this, this paper introduces RE-Searcher to explicitly integrate goal-oriented planning and self-reflection into the reasoning process. The method is first trained using SFT, followed by GRPO with a composite reward.

**Strengths:**

- The paper is well-written and easy to follow.
- The intuitive integration of goal-setting and binary self-reflection is easy to implement.
- Authors conduct comprehensive evaluation across divers datasets. Clear ablation studies shown the contribution of the reward design.

**Weaknesses:**

- The observed instability of success-rate in Sec. 2.1 may largely stem from the inherent differences in model capacity.
- In Sec. 2 and Sec. 4.2, authors primarily assess models by measuring the consistency of accuracy across multiple runs. However, in agentic search settings, the dominant source of instability stem from variations in search queries and inaccurate documents. The paper tackles the challenge with their search and reflection design, but lacks analysis of how the search queries changes before and after training, particularly whether the proposed goal-oriented planning and self-reflection lead to more consistent or robust query formulation.
-  According to my knowledge, the instruction employed in this work is not particularly challenging for a model like Qwen2.5-3B to follow. The necessity of warm-up needs further clarification.

**Questions:**

- The coefficient of the MBE reward is even smaller than that of the format reward. How does the model perform with larger MBE rewards?
- How does the model perform after the warm-up?

---

> ### Author Response · Authors · 2025-11-21
>
> # Model capacity and instability of success-rate (Weakness 1)
> We agree that model capacity can indeed influence the stability of search performance. In fact, capacity and stability are intertwined: stronger models are generally better at handling the variability of real-world search and tool-use, which tends to yield more stable trajectories.
>
> However, in Sec. 2.1 our goal is not to attribute the observed instability solely to capacity, but to highlight that such instability fundamentally limits the effective performance of search agents—even for reasonably strong models. As shown in Fig. 2, GPT-4o, which has substantial capacity and prior tool-use training, still exhibits a non-negligible “random right” portion, while smaller models such as Qwen2.5 (3B/7B) suffer from much more severe instability.
>
> The core contribution of our work is to mitigate this instability via training, so that smaller-capacity models can acquire robust search behaviors comparable to a much larger model. As illustrated in Fig. 6, after training, RE-Searcher based on Qwen2.5 (3B/7B) achieves “always right’’ and “random right’’ ratios that are on par with GPT-4o. This indicates that (1) instability is not merely a trivial artifact of model size, and (2) our method effectively transfers robust search policies from a strong model to smaller ones, substantially improving their stability and success rate.
>
> # Improvement of search query robustness (Weakness 2)
> Thank you for emphasizing the importance of query stability in agentic search settings. We agree that variations in search queries and noisy retrieval are key sources of instability, and our method is designed to mitigate them.
>
> Qualitatively, the response case in Fig.4 (right) illustrates that after training, the model can detect mismatches between retrieved documents and the search goal, reflect on failure, and revise its search keywords to better align with the intended objective. thereby obtaining the correct answer after a few iterations. This shows that goal-oriented planning and self-reflection lead to more targeted and consistent query reformulation compared to the untrained model.
>
> Quantitatively, Figure 7 reports a perturbation experiment where we deliberately distort the model’s search queries so that retrieval deviates from its original intent. We then measure performance degradation under these perturbed queries. Our model exhibits robustness comparable to GPT-4o, indicating a stable search policy that can recover from suboptimal or noisy queries.
> Overall, these results show that our design not only improves final accuracy, but also makes query formulation more consistent and robust.
>
> # Necessity of warm-up (Weakness 3 & Question 2)
> To verify whether models like Qwen2.5-3B can follow our relatively simple instruction template without additional warm-up, we conducted an ablation experiment in which the supervised warm-up phase was removed and the model was trained directly using reinforcement learning (“no warm-up”).
>
> In the no-warm-up setting, the format reward remained consistently low across RL training, indicating that the model failed to reliably follow the predefined response template. As shown in the table below, the format reward of the no-warm-up model plateaus around 0.09–0.10, whereas the model trained with the warm-up stage reaches around 0.38–0.44:
>
> | Training Step | 5 | 10 | 20  | 30 | 40 | 50 | 60 |
> |---------------|---|----|-----|----|----|----|----|
> |  w/o warm-up  | 0.0938 | 0.0959 | 0.0977  | 0.0998 | 0.0998 | 0.0990 | 0.0996 |
> |   w/ warm-up  | 0.384 | 0.415 | 0.411  | 0.395 | 0.440 | 0.388 | 0.387 |
>
> These observations suggest that, in our RL setup, the model’s generic instruction-following ability is not sufficient to guarantee stable learning of the specific formatting constraints. The warm-up stage provides the model with explicit supervised signals on the target template, which (i) significantly improves the format reward and (ii) leads to more stable and reliable RL training. Therefore, the warm-up is empirically necessary in our framework, even for a relatively capable base model like Qwen2.5-3B.

---

> ### Author Response · Authors · 2025-11-21
>
> # Sensitivity analysis of reward coefficient (Question 1)
>
> We initially set the reflection reward weight to 0.1 based on preliminary experiments and domain experience, aiming to balance the reflection reward against the other reward components. To address the reviewer’s concern more systematically, we have now conducted a sensitivity analysis over different coefficients. The results are summarized below:
>
> | coefficient | In domain | Out of domain | AVG.  |
> |------------|-----------|---------------|-------|
> | 0          | 0.4404    | 0.4204        | 0.4261 |
> | 0.1        | 0.4415    | **0.4465**    | **0.4451** |
> | 0.3        | 0.4360    | 0.4248        | 0.4280 |
> | 0.5        | **0.4425**| 0.4398        | 0.4406 |
>
> The results show that using 0.1 achieves the best overall performance in terms of the average score, and especially improves out-of-domain performance. In addition, all non-zero coefficients (0.1, 0.3, 0.5) outperform the setting without a reflection reward (0.0), which indicates that our method is not overly sensitive to the exact value of this weight and that the reflection reward consistently brings benefits. We will include this sensitivity study in the final version of the paper.

---

> ### Author Response · Authors · 2025-11-21
>
> Thank you very much for these constructive suggestions, which have significantly enriched the content and perspective of our paper. Below, we summarize several key comments from other reviewers and our corresponding responses. These points were also highly beneficial to our work and may be of interest to you:
>
> - **Dependency on external LLMs (Reviewer 4Xdc)**: Although we use GPT-4o-mini to supervise the reflection signal, this supervision is only applied to the final reflection output of the model (i.e., the <reflect>True/False</reflect> tokens). It only judges whether the reflection is correct, and does not involve any leakage of external knowledge. We further added a rule-based reward experiment, which demonstrates that the performance gains of RE-Searcher mainly come from methodological improvements rather than knowledge leakage from a more powerful model.
> - **Heterogeneous improvements on different datasets (Reviewers 4Xdc and i3qY)**: In Table 2, the performance gains of our model vary across datasets. This is mainly because NQ, TriviaQA, and PopQA are single-hop datasets: one search is usually sufficient to obtain relevant information, so the demand for sophisticated reflection behavior is relatively low. Consequently, our improvements on these datasets are less pronounced than on multi-hop datasets. In contrast, ZeroSearch employs a model-based search engine with stronger semantic matching capabilities and therefore performs better on single-hop data. Nonetheless, our approach still achieves competitive performance on single-hop datasets and attains SOTA results on average across all benchmarks.
> - **Computational cost of agent inference (Reviewers 4Xdc and tB4e)**: We report the number of output tokens and search steps for RE-Searcher and Search-R1. Our method indeed leads to a higher average number of generated tokens and search steps. However, an average of around 330 output tokens and about 3 search steps per query remains practically acceptable. Moreover, our method improves the stability of the model, making its behavior more reliable on some particularly challenging queries.
> - **Deeper explanation of the novelty (Reviewer tB4e)**: While planning (decomposing and organizing tasks) and reflection are common techniques in agent systems, our proposed goal planning is not a traditional task decomposition. Instead, it sets an explicit goal for the current action, akin to predicting the outcome of that action. Goal planning and reflection together form a prediction–action–reflection loop, enabling the model to interact and learn from the external environment in a manner similar to a world model, thus enhancing its robustness to environmental variability. We have added experiments comparing our approach with pure planning and pure reflection baselines, and the results confirm the advantages of our method.
> - **Application of process-level rewards (Reviewer i3qY)**: In our framework, the reflection reward is indeed defined at the process level, but the GRPO algorithm estimates a single episode-level advantage via group-wise regularization to reduce variance. This makes it impractical to incorporate step-wise supervision directly into GRPO, so we aggregate the reflection reward into the outcome reward in Eq. (3), and the resulting episode-level advantage is then broadcast to all action tokens in the trajectory. We also tried a more explicit but heuristic scheme that selectively upweighted the advantages of reflection tokens, i.e., $A_{reflect} := (1 + β · r) · A$. However, this breaks the variance-control design of GRPO and leads to instability, with RL training collapsing after about 20 steps. Based on these observations, we adopted the current design (adding reflection rewards to the outcome reward), which is significantly more stable and achieves better performance in our experiments.
>
> Once again, we sincerely thank all reviewers for their insightful comments. They have provided us with a more comprehensive view of our work and helped us refine and deepen our study. We will incorporate these new analyses and results into the final version of our manuscript.

---

> ### Author Response · Authors · 2025-11-27
>
> Dear Reviewer SaqS,
>
> We sincerely appreciate your valuable feedback on the paper. Given that the deadline for the discussion phase is approaching, we would be most grateful if you could kindly share any remaining concerns at your earliest convenience, and we will address them immediately.

---

### Official Review · Reviewer_4Xdc · 2025-11-01

**Soundness:** 3
**Presentation:** 3
**Contribution:** 2
**Rating:** 4
**Confidence:** 3

**Summary:**

The paper studies a real and under-discussed problem in agentic / search-augmented LLMs: small, plausible variations in search queries can push the agent onto a bad trajectory and the agent often fails to recover. The authors first quantify this “search fragility” by showing (i) high “random-right” rates across two runs and (ii) large drops in similarity after tiny query perturbations. To address this, they propose RE-Searcher, an agent that (1) explicitly states a search goal before issuing a query, (2) reflects on whether retrieved evidence satisfies that goal, and (3) is RL-trained (GRPO) with a mixed reward (format + factual + LLM-as-judge reflection) to keep the model in the “plan–search–reflect–answer” loop.

**Strengths:**

1. The work identifies a concrete failure mode of search agents, namely that a slightly different but still reasonable query can lead to very different retrieved evidence, which in turn damages answer quality.
2. The proposed template, “state goal → search → reflect → continue or answer,” is simple, explicit, and can be plugged into existing search-RL systems without complicated architecture changes.
3. Experiments on both 3B and 7B models, on in-domain and out-of-domain QA, plus ablations on the reward components, make the paper believable.

**Weaknesses:**

1. Training uses GPT-4o-mini (or a similar model) to score reflection quality. This supervision is non-trivial. It is unclear how much of the final robustness comes from this stronger teacher, rather than from the agent’s own structure.
2. The method may require several search steps, longer contexts, and an external LLM during training. At inference time, does the model always reflect, or can it stop early? A comparison with Search-R1 using the same search budget would be useful.

**Questions:**

1. Why was 0.1 chosen as the reflection reward weight? Was any systematic hyperparameter search conducted?

2. Can you propose or experiment with a self-supervised reflection mechanism that avoids dependency on external LLMs?

3. Why are RE-Searcher’s improvements smaller on some datasets (e.g., NQ) compared to others?

4. How much does the method increase computational cost relative to baselines? How many additional LLM calls per query are required?

---

> ### Author Response · Authors · 2025-11-21
>
> # Reward dependency on external LLMs (Weakness 1 & Question 2)
> We do use GPT-4o-mini to supervise the reflection quality when training RE-Searcher, but this supervision is very limited: GPT-4o-mini is only used to decide whether the model’s final reflection decision (i.e., the `<reflect>True/False</reflect>` tokens) is correct. It does not produce revised search trajectories or intermediate reasoning steps, nor does it indicate “what is wrong” or “how to fix it.” Therefore, it only provides a binary correctness signal, rather than distilling its own complex reasoning abilities into RE-Searcher.
>
> To further verify that the robustness of the model comes from our proposed agent structure itself, rather than relying on a stronger teacher, we construct a **fully LLM-free rule-based reflection supervision signal**. Specifically:
>
> - For HotpotQA, we use the annotated supporting facts as the evidence for the answer; for single-hop NQ, we use the final answer text as an approximation to supporting facts.
> - When the search agent retrieves content that contains these supporting facts, we label this retrieval as effective and set the ground-truth reflection label to True; otherwise, it is set to False.
> - Comparing this rule-based ground-truth reflection label with the model’s own `<reflect>True/False</reflect>` prediction yields the reflection reward.
>
> The experimental results under these two supervision strategies are shown below and are close:
>
> |                      | In domain | Out of domain | AVG.   |
> |----------------------|----------:|--------------:|-------:|
> | No reflection reward | 0.4404    | 0.4204        | 0.4261 |
> | Rule-based reward    | **0.4440** | 0.4376       | 0.4390 |
> | Model-based reward   | 0.4410    | **0.4465**    | **0.4450** |
>
> Note that the supporting facts are span-level text and may still have a semantic gap with the retrieved passages; on the other hand, the supporting-fact annotations in HotpotQA are known to contain noise. Even so, rule-based reward still offers a certain improvement over no reflection reward, and its effect is comparable to that of model-based reward. This indicates that:
> - GPT-4o-mini does not transfer high-level reasoning knowledge to the model here; it only provides a binary label of whether the reflection is correct.
> - The robustness of our method mainly comes from the agent structure and reflection mechanism of RE-Searcher itself, rather than from the reasoning ability of a stronger teacher.
>
> # Increase of computational cost (Weakness 2 & Question 4)
> Our method indeed introduces planning and reflection behaviors during training, which can lead to more search steps. We report the statistics of the number of search steps for RE-Searcher and Search-R1 in the table below. While planning and reflection do increase the average number of searches, the average remains around 3 steps with a smaller variance. The maximum number of search steps is 7, which is a reasonable cost for the search tasks.
>
> | Model              | average    | std        | max | min |
> |--------------------|-----------:|-----------:|----:|----:|
> | RE-Searcher-7B     | 3.20 | 1.08 | 7  | 1  |
> | Search-R1-instruct-7B | 1.72 | 1.31 | 5  | 1  |
>
> Moreover, at inference time, the model does not always perform multiple rounds of reflection. Our search procedure allows the model to stop early once it reaches a sufficiently confident answer, so for many easy instances the model conducts only a few searches (see the “min” column). This shows that RE-Searcher adaptively controls the search depth instead of always using the maximum search budget.
>
> As for additional LLM calls, these only occur during the model's training phase. During the inference phase, the model **relies entirely on its own capabilities for reflection** and no longer requires external supervision signals.

---

> ### Author Response · Authors · 2025-11-21
>
> # Sensitivity analysis of reward coefficient (Question 1)
> We initially set the reflection reward weight to 0.1 based on preliminary experiments and domain experience, aiming to balance the reflection reward against the other reward components. To address the reviewer’s concern more systematically, we have now conducted a sensitivity analysis over different coefficients. The results are summarized below:
>
> | coefficient | In domain | Out of domain | AVG.  |
> |------------|-----------|---------------|-------|
> | 0          | 0.4404    | 0.4204        | 0.4261 |
> | 0.1        | 0.4415    | **0.4465**    | **0.4451** |
> | 0.3        | 0.4360    | 0.4248        | 0.4280 |
> | 0.5        | **0.4425**| 0.4398        | 0.4406 |
>
> The results show that using 0.1 achieves the best overall performance in terms of the average score, and especially improves out-of-domain performance. In addition, all non-zero coefficients (0.1, 0.3, 0.5) outperform the setting without a reflection reward (0.0), which indicates that our method is not overly sensitive to the exact value of this weight and that the reflection reward consistently brings benefits. We will include this sensitivity study in the final version of the paper.
>
> # Heterogeneous improvements on different datasets (Question 3)
>
> We appreciate the reviewer’s careful observation. The heterogeneous gains across datasets are mainly due to the different reasoning requirements of each benchmark.
>
> First, datasets such as NQ, TriviaQA, and PopQA are predominantly **single-hop** QA tasks. Correct answers can often be obtained by **retrieving one relevant passage** (The search keywords may be contained within the question.) and performing shallow reasoning, so **they put relatively limited emphasis on complex reasoning and reflection**. Since RE-Searcher is specifically designed to improve iterative reasoning and reflection over multiple search steps, its advantage is naturally less pronounced on these simpler single-hop datasets than on multi-hop benchmarks such as HotpotQA, 2Wiki, Musique, and Bamboogle.
>
> Second, even on these single-hop datasets, our method performs competitively: RE-Searcher is slightly worse than ZeroSearch but remains comparable to or better than most other baselines (see Table 2). **The gap to ZeroSearch is due to its use of LLM as a simulated search engine**. ZeroSearch finetuned an LLM to directly predict search queries and their corresponding passages, and then uses this simulation LLM to replace the external search engine. Compared with a standard retrieval engine, such a simulation LLM has much stronger semantic matching ability and can often “retrieve” highly relevant information in one step. This design gives ZeroSearch a particular advantage on single-hop QA, where semantic matching is usually sufficient.
>
> However, on multi-hop datasets, strong semantic matching alone is not enough. The key challenge is to decompose the question into intermediate sub-queries and iteratively update them based on previous evidence. In this setting, our explicit reasoning–reflection loop is crucial for forming better search queries and using retrieved information more effectively. Consequently, RE-Searcher consistently outperforms all baselines, including ZeroSearch, on multi-hop benchmarks, leading to the best average performance across all datasets.

---

> ### Author Response · Authors · 2025-11-21
>
> Thank you very much for these constructive suggestions, which have significantly enriched the content and perspective of our paper. Below, we summarize several key comments from other reviewers and the abstracts of our corresponding responses. These points were also highly beneficial to our work and may be of interest to you:
>
> - **Deeper explanation of the novelty (Reviewer tB4e)**: While planning (decomposing and organizing tasks) and reflection are common techniques in agent systems, our proposed goal planning is not a traditional task decomposition. Instead, it sets an explicit goal for the current action, akin to predicting the outcome of that action. Goal planning and reflection together form a prediction–action–reflection loop, enabling the model to interact and learn from the external environment in a manner similar to a world model, thus enhancing its robustness to environmental variability. We have added experiments comparing our approach with pure planning and pure reflection baselines, and the results confirm the advantages of our method.
> - **Application of process-level rewards (Reviewer i3qY)**: In our framework, the reflection reward is indeed defined at the process level, but the GRPO algorithm estimates a single episode-level advantage via group-wise regularization to reduce variance. This makes it impractical to incorporate step-wise supervision directly into GRPO, so we aggregate the reflection reward into the outcome reward in Eq. (3), and the resulting episode-level advantage is then broadcast to all action tokens in the trajectory. We also tried a more explicit but heuristic scheme that selectively upweighted the advantages of reflection tokens, i.e., $A_{reflect} := (1 + β · r) · A$. However, this breaks the variance-control design of GRPO and leads to instability, with RL training collapsing after about 20 steps. Based on these observations, we adopted the current design (adding reflection rewards to the outcome reward), which is significantly more stable and achieves better performance in our experiments.
>
> Once again, we sincerely thank all reviewers for their insightful comments. They have provided us with a more comprehensive view of our work and helped us refine and deepen our study. We will incorporate these new analyses and results into the final version of our manuscript.

---

> ### Comment · Reviewer_4Xdc · 2025-11-23
>
> Thanks for the reply and additional experiments. I have a few additional questions, and I would greatly appreciate it if you could address them.
> 1. The fact that the simple keyword-matching rule yields performance comparable to that of the sophisticated GPT-4o-mini judge raises questions of whether the LLM-as-a-judge design is actually necessary.  Additionally, considering the computational cost of an external LLM during training, what motivates you to retain the complex model-based supervision instead of simple heuristics in your final proposed framework?
> 2. RE-Searcher almost doubles the average step count of Search-R1, 1.72 vs. 3.20. However, 'step count' is merely a proxy for cost. The approach adds significant computational overhead by generating \<think\>, \<goal\>, and \<reflect\> tags at each step. Could the authors provide the comparison of average total inference tokens per query?(An estimate is also acceptable.)
> 3. The lower gains on datasets like NQ are attributed to their simplicity, which favors semantic matching. Fair enough, but this suggests a potential weakness—the agent seems vulnerable to “over-planning” or “over-thinking” on simple queries for which deep reflection is unnecessary. A  robust agent would ideally recognize query complexity and adjust its search depth as appropriate. Does RE-Searcher have an adaptive mechanism that would bypass the expensive planning loop for simple tasks, or does it rigidly apply the full procedure regardless of query difficulty? If the latter, how would this impact efficiency on mixed-difficulty real-world workloads?

---

> > ### Author Response · Authors · 2025-11-24
> >
> > # Reasons for choosing model-based reward
> >
> > We sincerely thank the reviewer for this insightful question. It has prompted us to think more deeply about the core contribution of our work. We offer the following points in response:
> >
> > 1. **Convenience and Scalability of the LLM Judge**: While it may seem counterintuitive, leveraging an off-the-shelf model like GPT-4o-mini is often more convenient and scalable for developers. It requires minimal data preprocessing and no hand-crafting of rules; a well-designed prompt is sufficient to obtain immediate reward feedback. This ease of implementation was our primary motivation for adopting the LLM-as-a-judge approach in our initial design.
> > 2. **Focus on the Robustness of Reflection-based Reward**: Our primary contribution is the concept of a *reflection-based reward signal*, not the specific tool used to generate it. The fact that both a sophisticated LLM and a simple rule-based method yield comparable performance is a testament to the robustness of our proposed reward mechanism. It demonstrates that the core idea is sound and effective across different supervision sources.
> > 3. **Acknowledging the Value of the Heuristic Approach**: We agree that the rule-based method is a valuable and efficient baseline. Your comment inspired us to include this comparison, which has significantly strengthened our paper. It provides a meaningful ablation study that makes our research more comprehensive by showing that the benefits of our framework are not solely dependent on a powerful, costly LLM.
> >
> > In summary, while the rule-based approach is a strong and efficient alternative, the LLM-as-a-judge design remains a highly practical, flexible, and developer-friendly choice. The comparable performance of both methods reinforces our central claim about the effectiveness and robustness of the reflection-based reward.
> >
> > # Statistics of output tokens
> > We thank the reviewer for raising the concern about increased computational cost. We agree that explicit goal planning and reflection introduce additional tokens. The output token statistics are:
> >
> > | Model         | AVG.  | Std Dev | Max | Min |
> > |---------------|-------|--------|-----|-----|
> > | RE-Searcher-7B | 330.46 | 111.47 | 906 | 73  |
> > | Search-R1-7B   | 186.23 | 156.33 | 1331 | 48 |
> >
> > RE-Searcher indeed has a higher average output length, but ≈330 tokens remain practical for agent-style search. At the same time, it has a **smaller variance** and a **lower maximum length**, indicating more stable and predictable behavior. In difficult cases, the goal-planning and reflection steps help the model converge more quickly instead of repeatedly making unsuccessful attempts, mitigating extremely long trajectories.
> >
> > # Adaptive strategy in different situation
> >
> > We thank the reviewer for raising the concern about potential “over-planning” on simple queries and its impact on mixed-difficulty, real-world workloads. Our **goal-planning + self-reflection** mechanism is explicitly designed to make the search depth adaptive to query complexity. For each query, RE-Searcher first generates an explicit **search goal**. During the search process, the agent continuously checks whether the currently retrieved evidence is sufficient to satisfy this goal. Once it judges the goal to be met, the agent outputs `<reflect>True</reflect>` and **terminates the planning–search loop**, thereby avoiding unnecessary additional steps on simple questions. We report the number of search iterations on datasets of different complexity:
> >
> > |           | AVG. | Std Dev | Max | Min |
> > |----------|------|-----|-----|-----|
> > | Single-hop | 1.87 | 0.94 | 5   | 1   |
> > | Multi-hop  | 4.20 | 1.36 | 7   | 1   |
> > | Mixed      | 3.20 | 1.08 | 7   | 1   |
> >
> > As the table shows, on **single-hop (simpler) queries**, RE-Searcher typically performs fewer than 2 search iterations on average and often stops after a single retrieval round, indicating that it does not “over-think” when the task is simple. On **multi-hop (more complex)** queries, the agent automatically increases the number of search steps through self-reflection, enabling deeper reasoning and evidence gathering when needed. On mixed datasets, RE-Searcher adjusts its search depth between these regimes.
> >
> > In **real-world, mixed-difficulty workloads**, this behavior means that simple, factoid-style queries usually incur only 1–2 retrieval rounds, while only genuinely complex queries trigger deeper planning and multiple search iterations. Thus, RE-Searcher does not rigidly apply a fixed, expensive procedure to all queries; instead, its goal-planning and self-reflection mechanism provides **adaptive control of search depth**, which keeps computational cost manageable without sacrificing performance on harder questions.

---

> > > ### Comment · Reviewer_4Xdc · 2025-11-24
> > >
> > > Thank you for the response, I will be raising my final rating accordingly.

---

> > > > ### Author Response · Authors · 2025-11-24
> > > >
> > > > We are very grateful to you for the constructive comments and for the decision to raise the score. The suggestions provided were instrumental in improving our work, and we welcome any further dialogue.

---

### Author Response · Authors · 2025-11-26

We would like to sincerely thank the reviewers for their constructive comments. Based on these suggestions, we have conducted additional experiments to make our research more comprehensive and in-depth. We have incorporated these results into the revised manuscript and uploaded it to the system. All changes have been highlighted in **red**.

We hope that the revised manuscript addresses your concerns regarding the overall presentation and the depth of our research. If there are any further concerns, please let us know, and we will be happy to address them. If you find our revisions satisfactory and feel our work is sufficiently improved, we would greatly appreciate your support in reconsidering the score.

---

### Author Response · Authors · 2025-12-02
**Summary of Rebuttal during ICLR Incident [1/3]**

Dear Area Chairs,

Given this unexpected incident, we fully understand the challenges the ICLR organizing committee is facing and the additional responsibilities you have taken on. To make the work easier, we have summarized our main contributions, central rebuttal points, and the progression of review scores. Further details are presented below.

## Summary of Our Paper

As large language models (LLMs) continue to improve on logical reasoning and knowledge-intensive question answering, it has become a clear trend to endow *agentic* models with richer tool-use capabilities and to deploy them in realistic environments, rather than restricting them to simple QA settings. However, real-world environments bring not only additional capabilities, but also substantially greater uncertainty. In this work, we focus on **search agents** as a concrete instantiation, and study the potential issues that arise when agentic models interact with complex environments. We further propose a simple yet effective method to mitigate these problems:

- We introduce a novel framework to qualitatively and quantitatively evaluate the **diversity and complexity of the environment**, and find that even small perturbations during the interaction process can be amplified by the environment, eventually leading to larger errors. This reveals that agent–environment interaction requires a more robust mechanism in order to avoid such risks.
- We design an **agentic framework** that integrates **goal-oriented planning** and **self-reflection**, enabling the agent to explicitly predict a concrete goal for each action it is about to take and then reflect on whether the action outcome matches the predicted goal. This *predict–act–reflect* behavior pattern allows the model to gradually form an internal understanding of the environment (by comparing its predicted outcomes with the actual results and adjusting its predictions accordingly), and thereby make better interaction decisions.
- Extensive experiments show that our method not only improves the performance of search agents across multiple search benchmarks, but also makes them more resilient to external perturbations from the environment, leading to more robust search behavior.

## The Strengths Summarized by Reviewers

- **Clear Motivation & Problem Identification**: The paper clearly identifies and analyzes a concrete failure mode / instability of search agents (environmental variability, output stochasticity, fragility of search queries), with preliminary experiments that empirically demonstrate the problem.(Reviewer 4Xdc, tB4e, i3qY)
- **Simple & Intuitive Method**: The proposed method is simple, intuitive, and clearly described: a structured generation loop (“state goal → search → reflect → continue or answer”) that extends the agent loop with goal-setting and binary self-reflection, and can be plugged into existing search-RL systems without complex architecture changes.(Reviewer 4Xdc, SaqS, tB4e, i3qY)
- **Well-Written & Easy to Follow**: The paper is well-written, clearly structured, and easy to follow, with methodological descriptions and figures that are easy to understand.(Reviewer SaqS, tB4e, i3qY)
- **Comprehensive & Strong Experiments**: The experimental evaluation is comprehensive and convincing: it covers multiple model scales (3B and 7B), in-domain and out-of-domain QA, and a wide range of baselines, achieving strong overall performance.(Reviewer 4Xdc, SaqS, i3qY)
- **Thorough Ablation & Robustness Analysis**: The paper includes clear and informative ablation and robustness analyses (including reward component ablations and robustness/Pass@2 analyses) that highlight the contribution of the reward design and show that the method substantially improves robustness (e.g., reduces the random-right ratio).(Reviewer 4Xdc, SaqS, tB4e, i3qY)

---

> ### Author Response · Authors · 2025-12-02
> **Summary of Rebuttal during ICLR Incident [2/3]**
>
> ## Main Response to Reviewers' Comments
>
> We have summarized the reviewers' main comments and our corresponding responses below. More detailed comments have also been addressed with additional experiments and replies, and the corresponding revisions have been made in the manuscript PDF. All modifications are highlighted in red font for your convenience.
>
> > Q1: The paper uses GPT-4o-mini to supervise the reflection reward, which may cause knowledge leakage (Reviewer 4Xdc).
>
> A1: Although we use GPT-4o-mini to supervise the reflection signal, this supervision is only applied to the model’s final reflection output (i.e., the `<reflect>True/</reflect>` tokens). It only evaluates whether the reflection is correct and does not introduce or leak any external knowledge.
>
> We further conducted an experiment using a purely rule-based reward:
>
> |                      | In domain | Out of domain | AVG.   |
> |----------------------|----------:|--------------:|-------:|
> | No reflection reward | 0.4404    | 0.4204        | 0.4261 |
> | Rule-based reward    | **0.4440** | 0.4376       | 0.4390 |
> | Model-based reward   | 0.4410    | **0.4465**    | **0.4450** |
>
> The results show that the rule-based approach achieves comparable performance, indicating that the performance gains come from the effectiveness of our method itself rather than information leakage from the external model.
>
> > Q2: Does the use of goal-setting and self-reflection introduce additional computational overhead, and does it cause the model to overthink? (Reviewers 4Xdc and tB4e)
>
> A2: We report the number of output tokens and search steps for RE-Searcher and Search-R1 in Tables 1 and 2 below. Our method does lead to a higher average number of generated tokens and search steps. However, an average of around 330 output tokens and about 3 search steps per query remains practically acceptable. In addition, our method improves model stability, making its behavior more reliable on particularly challenging queries.
>
> **Table 1: Statistics of search steps**
> | Model                  | Average | Std Dev | Max | Min |
> |------------------------|--------:|--------:|----:|----:|
> | RE-Searcher-7B         |   3.20  |   1.08  |  7  |  1  |
> | Search-R1-instruct-7B  |   1.72  |   1.31  |  5  |  1  |
>
> **Table 2: Statistics of output tokens**
> | Model           | Average | Std Dev | Max  | Min |
> |-----------------|--------:|--------:|-----:|----:|
> | RE-Searcher-7B  |  330.46 |  111.47 |  906 |  73 |
> | Search-R1-7B    |  186.23 |  156.33 | 1331 |  48 |
>
> **Table 3: Statistics of search steps on different datasets**
> | Dataset Type | Average | Std Dev | Max | Min |
> |-------------|--------:|--------:|----:|----:|
> | Single-hop  |   1.87  |   0.94  |  5  |  1  |
> | Multi-hop   |   4.20  |   1.36  |  7  |  1  |
> | Mixed       |   3.20  |   1.08  |  7  |  1  |
>
> Moreover, Table 3 reports the statistics of search steps for RE-Searcher on datasets of different types. The results show that our method enables the model to adapt its reasoning strategy to the difficulty of the query and perform adaptive search accordingly.
>
> > Q3: Why is the coefficient of the reflection reward in Equation (3) set to 0.1? (Reviewers 4Xdc, SaqS, and i3qY)
>
> A3: We initially set the reflection reward coefficient to 0.1 based on empirical considerations, aiming to balance the contributions of different reward components. We have now conducted a sensitivity analysis on this coefficient and report the results in the table below. The results support our original choice.
>
> | Coefficient | In-domain | Out-of-domain | Avg.   |
> |------------|-----------|---------------|--------|
> | 0          | 0.4404    | 0.4204        | 0.4261 |
> | 0.1        | 0.4415    | **0.4465**    | **0.4451** |
> | 0.3        | 0.4360    | 0.4248        | 0.4280 |
> | 0.5        | **0.4425**| 0.4398        | 0.4406 |
>
> > Q4: In Table 2, why are the performance improvements of the model not as significant on some datasets? (Reviewers 4Xdc and i3qY):
>
> In Table 2, the performance gains of our model vary across datasets. This is mainly because NQ, TriviaQA, and PopQA are single-hop datasets, where a single search is usually sufficient to retrieve the necessary information. As a result, the need for sophisticated reflection mechanisms is relatively limited, and our improvements on these datasets are therefore less pronounced than on multi-hop datasets.
>
> By contrast, ZeroSearch employs a model-based search engine with stronger semantic matching capabilities, which allows it to perform particularly well on single-hop data. Nevertheless, our approach still achieves competitive performance on these single-hop datasets and delivers state-of-the-art results on average across all benchmarks.

---

> ### Author Response · Authors · 2025-12-02
> **Summary of Rebuttal during ICLR Incident [3/3]**
>
> > Q5: The novelty of core concepts is limited. The core ideas of "planning" (decomposing a problem) and "reflection" (evaluating retrieved information) are well-established concepts in agentic AI and RAG literature. (Reviewer tB4e)
>
> While planning (decomposing and organizing tasks) and reflection are common techniques in agent systems, our proposed goal planning is not a traditional task decomposition. Instead, it sets an explicit goal for the current action, akin to predicting the outcome of that action. Goal planning and reflection together form a prediction–action–reflection loop, enabling the model to interact and learn from the external environment in a manner similar to a world model, thus enhancing its robustness to environmental variability. We have added experiments comparing our approach with pure planning and pure reflection baselines, the results are shown below, which confirms the advantages of our method.
>
> **Table 1 comparison with pure planning and pure reflection**
>
> |                      | In domain | Out of domain | AVG.  |
> |----------------------|----------:|-------------:|------:|
> | Pure planning        | 0.4061    | 0.4362       | 0.4339 |
> | Pure reflection      | 0.4275    | 0.4234       | 0.4246 |
> | RE-Searcher   | **0.4410**| **0.4465**   | **0.4450** |
>
> **Table 2 experiments of anti-disturbance**
>
> | Model            | Accuracy Rate (%) | Degration (%) |
> |------------------|-------------------|---------------|
> | GTP-4o           | 88.28             | 11.72         |
> | Search-R1 (7B)   | 78.70             | 21.30         |
> | RE-Searcher (7B) | 87.27             | 12.73         |
> | RE-Searcher (3B) | 83.40             | 16.60         |
> | Pure Planning    | 84.38             | 15.62         |
> | Pure Refleciton  | 82.64             | 17.36         |
>
> > Q6: In Equation (3), the reflection reward is added to the outcome reward. Why not instead treat it as a process reward? (Reviewer i3qY)
>
> In our framework, the reflection reward is indeed defined at the process level, but the GRPO algorithm estimates a single episode-level advantage via group-wise regularization to reduce variance. This makes it impractical to incorporate step-wise supervision directly into GRPO, so we aggregate the reflection reward into the outcome reward in Eq. (3), and the resulting episode-level advantage is then broadcast to all action tokens in the trajectory. We also tried a more explicit but heuristic scheme that selectively upweighted the advantages of reflection tokens, i.e., $A_{reflect} := (1 + β · r) · A$. However, this breaks the variance-control design of GRPO and leads to instability, with RL training collapsing after about 20 steps. Based on these observations, we adopted the current design (adding reflection rewards to the outcome reward), which is significantly more stable and achieves better performance in our experiments.
>
> ## Score Changes
> Our paper initially received the scores 4, 6, 4, and 4. After our response, all of reviewer 4Xdc’s concerns were fully addressed, and they raised their score from 4 to 6. Reviewer i3qY stated in their initial comments that they would adjust their score based on our rebuttal; however, we have not received any further response from them. Similarly, reviewers tB4e and SaqS have not responded to our rebuttal.
>
> All of the above communication and score changes took place before November 27. We also confirm that we have never used any system vulnerabilities, nor have we engaged in any non-public communication with any of the reviewers.
>
> We acknowledge that the circumstances surrounding this incident were highly unexpected. We sincerely appreciate the challenges the ICLR organizing committee is navigating and recognize the additional burden placed upon the newly assigned Area Chairs. We wish to affirm our strict adherence to all double-blind review policies throughout this process. It is our hope that the provided summary will help alleviate your workload. We respectfully ask for your consideration in accepting our work and remain dedicated to upholding the principles of fairness and contributing to the advancement of our academic community.

---

### Meta-Review · Area_Chair_ALru · 2025-12-30

**Summary:**

This paper studies search fragility in search-augmented LLMs and proposes RE-Searcher, a structured loop trained with GRPO and a composite reward. However, the core ideas (planning and reflection) are well-estabilished and the incremental novelty is limited. In addition, some key points (e.g., isolating the effect of goal-planning vs reflection, and cost- and budget-matched comparisons) were raised as important for confidence in the claims. Although one reviewer reported being satisfied after the rebuttal and raised their score, the broader discussion did not converge across reviewers. Thus, I recommend rejection.

**Reviewer Concerns:**

Several key issues remain or are not fully convincing given the current evidence: novelty and conceptual distinctiveness are still questioned (planning and reflection are widely used), and at least one reviewer noted the missing isolation of goal-planning vs reflection. In addition, computational cost concerns persist because the approach expands per-step generation, and reviewers asked for stricter budget-matched baselines

**Reviewer Scores:**

Since most of the concerns of the negative reviewers (SaqS and i3qY) are not well addressed yet, I believe all the reviewers would possibly remain at the same score.

---

### Decision · Program_Chairs · 2026-01-26

Reject